# Efficient and Private Marginal Reconstruction with Local Non-Negativity

**Brett Mullins**[1]    **Miguel Fuentes**[1]    **Yingtai Xiao**[2]    **Daniel Kifer**[2]
**Cameron Musco**[1]    **Daniel Sheldon**[1]
[1]University of Massachusetts, Amherst    [2]Penn State University
{bmullins,mmfuentes,cmusco,sheldon}@cs.umass.edu
{yxx5224,duk17}@psu.edu

## Abstract

Differential privacy is the dominant standard for formal and quantifiable privacy and has been used in major deployments that impact millions of people. Many differentially private algorithms for query release and synthetic data contain steps that reconstruct answers to queries from answers to other queries that have been measured privately. Reconstruction is an important subproblem for such mechanisms to economize the privacy budget, minimize error on reconstructed answers, and allow for scalability to high-dimensional datasets. In this paper, we introduce a principled and efficient postprocessing method ReM (Residuals-to-Marginals) for reconstructing answers to marginal queries. Our method builds on recent work on efficient mechanisms for marginal query release, based on making measurements using a *residual query basis* that admits efficient pseudoinversion, which is an important primitive used in reconstruction. An extension GReM-LNN (Gaussian Residuals-to-Marginals with Local Non-negativity) reconstructs marginals under Gaussian noise satisfying consistency and non-negativity, which often reduces error on reconstructed answers. We demonstrate the utility of ReM and GReM-LNN by applying them to improve existing private query answering mechanisms.

## 1 Introduction

Differential privacy is the dominant standard for formal and quantifiable privacy and has been used in major deployments that impact millions of people such as the 2020 US Decennial Census [1]. One of the most fundamental problems in differential privacy is answering a workload of linear queries. Linear queries are used for basic descriptive statistics like counts and sums, and as building blocks for more complex tasks. Marginal queries, which describe the frequency distribution of subsets of discrete variables (e.g., income by age and education), are of particular interest as descriptive statistics and for use in downstream tasks like regression analyses.

A key subproblem in linear query answering is *reconstruction*. Given a workload of linear queries, most mechanisms select a different set of queries to measure to make the most efficient use of the privacy budget, and then use the noisy answers to reconstruct answers to workload queries [2–11]. Effective reconstruction methods can combine information from all noisy measurements to provide mutually consistent answers to workload queries.

Computational complexity is a key challenge for reconstruction methods. These methods answer workload queries by—either explicitly or implicitly—reconstructing a data distribution that has size exponential in the number of variables. To scale to high-dimensional data sets, existing approaches must represent this distribution compactly through some form of parametric representation [8–12], which introduces tradeoffs such as a restricted space of data distributions that can be represented [8–

38th Conference on Neural Information Processing Systems (NeurIPS 2024).

11], non-convex optimization objectives to find the best representation [9–11], or complexity that depends on the measured queries and is still exponential in the worst case [12].

We introduce ReM (residuals-to-marginals), a principled and scalable post-processing method to reconstruct answers to a workload of marginal queries from noisy measurements of *residuals*. Residuals are a class of linear queries that are related to marginals, which were recently introduced in the privacy literature [6] but previously studied in statistics [13, 14]. ReM uses a compact representation of the data distribution to produce workload answers without exponential complexity in the number of variables. ReM builds on the reconstruction approach of ResidualPlanner [6], which utilizes Kronecker structure to efficiently perform pseudoinverse operations. ReM is a flexible framework for performing reconstruction in a broad range of settings and it can be used with a variety of existing query-answering mechanisms. ReM also extends to the common setting of reconstructing answers to marginal queries from a set of noisy marginal measurements with isotropic Gaussian noise. In this case, ReM performs the standard pseudoinverse reconstruction and is the first method to do so efficiently. We also develop GReM-LNN (Gaussian ReM with local non-negativity), an extension that reconstructs marginals satisfying non-negativity, which often reduces error on reconstructed answers.

We demonstrate the utility of ReM and GReM-LNN by showing that they significantly reduce error and enhance the scalability of existing private query answering mechanisms including ResidualPlanner [6] and the multiplicative weights exponential mechanisms (MWEM) [15]. Our code is available at `https://github.com/bcmullins/efficient-marginal-reconstruction`.

## 2    Preliminaries

We consider a sensitive tabular dataset $\mathcal{D}$ of records $x^{(1)}, \ldots, x^{(N)}$. Each record $x = (x_1, \ldots, x_d)$ consists of $d$ categorical attributes. The $i$th attribute $x_i$ belongs to the finite set $\mathcal{X}_i$ of size $n_i$. The data universe is $\mathcal{X} = \prod_{i=1}^{d} \mathcal{X}_i$ and has size $n = \prod_i n_i$. The *data vector* or *data distribution* $p \in \mathbb{R}^n$ is a vector indexed by $\mathcal{X}$ that counts the occurrences of each record in $\mathcal{D}$; it has entries $p(x) = \sum_{i=1}^{N} \mathbb{I}[x^{(i)} = x]$. Since $n$ is exponential in the data dimension $d$, it is computationally intractable to work directly with data vectors in high dimensions.

### 2.1    Linear queries, marginals, and residuals

Linear queries are a rich class of statistics that include counts, sums, and averages and are used as building blocks for more complex tasks. A linear query is the sum of a real-valued function $q : \mathcal{X} \to \mathbb{R}$ applied to each record in the dataset. We adopt the equivalence that a query is a vector $q \in \mathbb{R}^n$ with answer $q^\top p$. A *query matrix* or *workload* $W$ is a collection of $m$ linear queries arranged row-wise in an $m \times n$ matrix. The answer to workload $W$ for data vector $p$ is given by $Wp$.

*Marginal queries* are a common type of linear query for high-dimensional data. They count the number of records that match certain values for a subset of the attributes – e.g., the number of people in a dataset with education at least a college degree and income \$50-\$100K. Let $\gamma \subseteq [d]$ be a subset of attributes and $x_\gamma = (x_i)_{i \in \gamma}$ be the corresponding subvector of a record $x$. Further, let $\mathcal{X}_\gamma = \prod_{i \in \gamma} \mathcal{X}_i$ and $n_\gamma = \prod_{i \in \gamma} n_i$. The *marginal* $\mu_\gamma \in \mathbb{R}^{n_\gamma}$ has entries $\mu_\gamma(t) = \sum_{i=1}^{N} \mathbb{I}[x_\gamma^{(i)} = t]$ that count the number of occurrences in the dataset for each setting $t \in \mathcal{X}_\gamma$ of the attributes in $\gamma$. Let $M_\gamma \in \mathbb{R}^{n_\gamma \times n}$ be the *marginal workload* so that $\mu_\gamma = M_\gamma p$. As shown in Fig. 1a, $M_\gamma$ can be written concisely as a Kronecker product over dimensions, with base matrices equal to the identity $I_k \in \mathbb{R}^{n_k \times n_k}$ for attributes in $\gamma$ and the all ones vector $1_k^\top \in \mathbb{R}^{1 \times n_k}$ for attributes not in $\gamma$. Kronecker product matrices can be understood as applying different linear operations along each dimension of a multi-dimensional array. In this case $M_\gamma$ sums over dimensions of the array representation of $p$ for attributes not in $\gamma$. We provide a brief summary of Kronecker products and their relevant properties in Appendix A.

*Residual queries* are class of linear queries closely related to marginals. They were recently introduced in the privacy literature [6] but previously studied in statistics as variable *interactions* [13, 14]. For $\tau \subseteq [d]$, the $\tau$-*residual* is obtained from the marginal $\mu_\tau$ by applying a differencing operator along each dimension. Let $D_{(k)}$ be the linear operator that computes successive differences for vectors of length $n_k$, i.e., $(D_{(k)}v)_i = v_{i+1} - v_i$ for $i = 1, \ldots, n_k - 1$; an example is shown for $n_k = 3$ in Fig. 1b. Let $D_\tau$ be the matrix that applies this operation to all attributes in the $\tau$-marginal as shown in

$$M_\gamma = \bigotimes_{k=1}^{d} \begin{cases} I_k & k \in \gamma \\ 1_k^\top & k \notin \gamma \end{cases} \qquad D_{(k)} = \begin{bmatrix} 1 & -1 & 0 \\ 0 & 1 & -1 \end{bmatrix} \qquad D_\tau = \bigotimes_{k=1}^{d} \begin{cases} D_{(k)} & k \in \tau \\ 1 & k \notin \tau \end{cases} \qquad R_\tau = \bigotimes_{k=1}^{d} \begin{cases} D_{(k)} & k \in \tau \\ 1_k^\top & k \notin \tau \end{cases}$$

(a) Marginals     (b) Differencing operator for $k$th attribute.     (c) Differencing operator for $\tau$-marginal.     (d) Residuals

Figure 1: Kronecker structure of workloads.

Fig. 1c. The residual workload can be written as $R_\tau = D_\tau M_\tau \in \mathbb{R}^{m_\tau \times n}$ where $m_\tau = \prod_{i \in \tau}(n_i - 1)$, which has the explicit Kroecker product form shown in Fig. 1d.[1] With these definitions, if $\mu_\tau = M_\tau p$ is the $\tau$-marginal, the $\tau$-residual is $\alpha_\tau = D_\tau \mu_\tau = R_\tau p$ and can be computed from either $\mu_\tau$ or $p$.

Residuals and marginals have an intricate structure. The $\gamma$-marginal is uniquely determined by the $\tau$-residuals for $\tau \subseteq \gamma$, i.e., there is an invertible linear transformation between $M_\gamma$ and $(R_\tau)_{\tau \subseteq \gamma}$ (a vertical block matrix). Intuitively, a $\gamma$-residual contains information *not* contained in the $\tau$-marginals for $\tau \subset \gamma$. Further, the row spaces of $R_\tau$ and $R_{\tau'}$ are orthogonal for any $\tau \neq \tau'$, and the row spaces of $M_\gamma$ and $R_\tau$ are orthogonal when $\tau \not\subseteq \gamma$ [6, 13, 14]. Along with Kronecker structure, the orthogonality of residuals is the key property we will leverage to perform efficient reconstruction.

A key advantage of residual workloads is that we can work with their pseudoinverses efficiently in certain situations even though they have exponential size. Let $Q^+$ denote the Moore-Penrose pseudoinverse of $Q$. The following proposition builds on the reconstruction method in [6] and will be used to reconstruct answers to a marginal query $M_\gamma$ from measurements for a collection of residuals.

**Proposition 1.** *Let $R_\mathcal{S} = (R_\tau)_{\tau \in \mathcal{S}}$ be a combined workload of residual queries for all $\tau$ in a collection $\mathcal{S} \subseteq 2^{[d]}$, where the individual matrices $R_\tau$ are stacked vertically. The size of $R_\mathcal{S}$ is $m \times n$ where $m = \sum_{\tau \in \mathcal{S}} m_\tau$. Then for any $z = (z_\tau)_{\tau \in \mathcal{S}} \in \mathbb{R}^m$ and any $\gamma$, it holds that*

$$M_\gamma R_\mathcal{S}^+ z = \sum_{\tau \in \mathcal{S}, \tau \subseteq \gamma} A_{\gamma,\tau} z_\tau, \qquad \text{where } A_{\gamma,\tau} := \bigotimes_{k=1}^{d} \begin{cases} D_{(k)}^+ & k \in \tau \\ (1/n_k)1_k & k \in \gamma \setminus \tau \\ 1 & k \notin \gamma \end{cases} \quad \text{for } \tau \subseteq \gamma.$$

*The matrix $A_{\gamma,\tau}$ has size $n_\gamma \times m_\tau$ and maps from the space of $\tau$-residuals to the space of $\gamma$-marginals. The running time to compute $A_{\gamma,\tau} z_\tau$ is $\mathcal{O}(|\gamma|n_\gamma)$.*

The proof of this result appears in Appendix C. The analysis of time complexity appears in Appendix E.

## 2.2 Differential Privacy

When releasing the results of any analysis performed on sensitive data, particular care needs to be taken to avoid leaking private information contained in the dataset. Differential privacy is a mathematical criterion that bounds the effect of any individual in the dataset on the output of a mechanism, which is satisfied by adding noise to the computation. This allows for formal quantification of the privacy risk associated with any release of information.

**Definition 1.** (Differential Privacy; [16]) Let $\mathcal{M} : \mathcal{X} \to \mathcal{Y}$ be a randomized mechanism. For any neighboring datasets $\mathcal{D}, \mathcal{D}'$ that differ by adding or removing at most one record, denoted $\mathcal{D} \sim \mathcal{D}'$, and all measurable subsets $S \subseteq \mathcal{Y}$: if $\Pr(\mathcal{M}(\mathcal{D}) \in S) \leq \exp(\epsilon) \cdot \Pr(\mathcal{M}(\mathcal{D}') \in S) + \delta$, then $\mathcal{M}$ satisfies $(\epsilon, \delta)$-approximate differential privacy, denoted $(\epsilon, \delta)$-DP.

A fundamental property of differential privacy relevant to our work is the post-processing property, which states that transformations of differentially private outputs that do not access the sensitive dataset $\mathcal{D}$ maintain their privacy guarantees. Formally:

**Proposition 2** (Post-processing; [17]). *Let $\mathcal{M}_1 : \mathcal{X} \to \mathcal{Y}$ satisfy $(\epsilon, \delta)$-DP and $f : \mathcal{Y} \to \mathcal{Z}$ be a randomized algorithm. Then $\mathcal{M} : \mathcal{X} \to \mathcal{Z} = f \circ \mathcal{M}_1$ satisfies $(\epsilon, \delta)$-DP.*

The reconstruction methods we propose in this paper are post-processing algorithms that take as input a set of noisy linear query answers and, thus, inherit the privacy guarantees from those noisy answers. Note that the present analysis is largely agnostic to the model of differential privacy used.

---

[1]Note that our matrix $D_\tau$ is slightly different from the operator used in [6] but has the same row space [14].

We discuss variants of differential privacy and privacy guarantees for query answering in Appendix B.

## 2.3 Private query answering

In private query answering, we are given a *workload* of linear queries $W \in \mathbb{R}^{m \times n}$. We seek to approximate the answers $Wp$ as accurately as possible while satisfying differential privacy. A general recipe for private query answering is *select-measure-reconstruct*. *Data-independent* mechanisms following this recipe such as the various matrix mechanisms [2–6] select and measure a set of queries $Q$ and reconstruct answers to $W$. *Data-dependent* mechanisms following this recipe such as MWEM [15] and various synthetic data mechanisms [7, 9, 10, 18, 19] typically maintain a model $\hat{p}$ of the data distribution $p$ that is improved iteratively by repeating the steps of select-measure-reconstruct and adaptively measuring queries that are poorly approximated by the current model $\hat{p}$. The key idea is that it is often possible to obtain lower error by measuring a different set of queries $Q$ than $W$ and then using answers to $Q$ to reconstruct answers for $W$. In this paper, we focus on the reconstruction subproblem and propose methods applicable to both the data-independent and data-dependent settings.

## 2.4 Query answer reconstruction

Reconstruction is a central subproblem to query answering. Suppose $y = Qp + \xi$ is the a set of measurements. To reconstruct a data distribution, we seek $\hat{p}$ such that $Q\hat{p} \approx y$. One method is to set $\hat{p} = Q^+ y$ where $Q^+$ is the Moore-Penrose pseudoinverse. This method is used in the matrix mechanism [4] and HDMM [5] but is not tractable in high dimensions. One contribution of our proposed method is to demonstrate that this pseudoinverse reconstruction is tractable when the query matrix $Q$ is a set of marginal measurements and $\xi$ is isotropic Gaussian noise. Other reconstruction methods such as Private-PGM [12] and those used by the mechanisms PrivBayes [8], GEM [9], RAP [10], and RAP++ [11] represent $\hat{p}$ through a parametric representation. These (usually) ensure tractability in high dimensions by using a compact representation, but introduce different tradeoffs. The parametric assumption typically restricts the space of data distributions that can be represented [8–11]. Optimizing over the parameteric representation is often non-convex, potentially leading to suboptimal optimization [9–11]. Private-PGM solves a convex optimization problem and is closest to the methods of this paper. However its complexity depends on the measured queries and is still exponential in the worst case [12]; our methods will not have exponential complexity.

We note that all of these above reconstruction methods, and the methods presented in this work, only depend on the dataset through the noisy query answers and, thus, satisfy the same degree of privacy as the answers by the post-processing property of differential privacy (Proposition 2).

## 3 Efficient Marginal Reconstruction from Residuals

In this section, we discuss methods for reconstructing answers to a workload of marginal queries given measurements of residuals. These methods utilize the structure of marginals and residuals to make reconstruction tractable and minimize error. Let $\mathcal{W} \subseteq 2^{[d]}$ and $M_{\mathcal{W}} = (M_\gamma)_{\gamma \in \mathcal{W}}$ be the combined workload of marginals for all of the attribute sets in $\mathcal{W}$ (e.g., all pairs or triples of attributes). Similarly, let $R_{\mathcal{S}} = (R_\tau)_{\tau \in \mathcal{S}}$ represent a set of residual queries for all $\tau$ in a collection $\mathcal{S}$. Our goal is to estimate the marginal query answers $M_{\mathcal{W}}p$ from noisy measurements $z = R_{\mathcal{S}}p + \xi$.

**ResidualPlanner.** ResidualPlanner [6] solves this problem elegantly in the matrix mechanism (i.e. data-independent) setting under Gaussian noise. Let $\mathcal{W}^{\downarrow} = \{\tau \subseteq \gamma : \gamma \in \mathcal{W}\}$ denote the *downward closure* of $\mathcal{W}$. When $\mathcal{S} = \mathcal{W}^{\downarrow}$, the residual queries for $\mathcal{S}$ uniquely determine

---

**Algorithm 1** ResidualPlanner reconstruction

---

**Input:** Marginal workload $\mathcal{W}$, $\mathcal{S} = \mathcal{W}^{\downarrow}$, measurements $z_\tau = R_\tau p + \mathcal{N}(0, \Sigma_\tau)$ for $\tau \in \mathcal{S}$
1: Reconstruct $\hat{\mu}_\gamma = \sum_{\tau \subseteq \gamma} A_{\gamma,\tau} z_\tau$ for $\gamma \in \mathcal{W}$

---

the marginals for $\mathcal{W}$, i.e., there is an invertible linear transformation between $M_{\mathcal{W}}$ and $R_{\mathcal{S}}$. This yields the reconstruction approach in Alg. 1. We suppose the residual queries $R_\tau$ are measured with Gaussian noise to yield $z_\tau$. In Line 1, the marginals are reconstructed by applying the invertible transformation from residuals to marginals. This reconstruction is equivalent to setting $\hat{\mu}_\gamma = M_\gamma \hat{p}$ where $\hat{p} = R_{\mathcal{S}}^+ z$ and $z = (z_\tau)_{\tau \in \mathcal{S}}$ by Proposition 1.

The full ResidualPlanner algorithm additionally chooses each $\Sigma_\tau = \sigma_\tau^2 D_\tau D_\tau^\top$ such that the resulting algorithm *optimally* answers the marginal workload indexed by $\mathcal{W}$ to minimize error under a natural class of convex loss functions for a given privacy budget [6]. That this can be done efficiently for a broad class of error metrics for marginal workloads is significant given the computational challenges that are often faced when attempting to optimally select measurements and reconstruct workload answers in high dimensions.

**A general approach to reconstruction.**    We propose a reconstruction algorithm that, like the one in ResidualPlanner, is efficient and principled, but that applies in more general settings. Reconstruction in ResidualPlanner uses the invertible transformation from residuals to marginals. This restricts to the case where the measured queries *exactly* determine the workload queries in the absence of noise. To address the full range of applications, it is important to address the cases where workload queries are overdetermined, underdetermined, or both.

Our proposed algorithm, ReM, is shown in Alg. 2. Compared to ResidualPlanner, the main differences are: (1) the set $\mathcal{S}$ of measured residuals is arbitrary, (2) a residual query can be measured any number of times with any noise distribution, (3) an optimization problem is solved for each $\tau$ to estimate the true residual query answer $\hat{\alpha}_\tau \approx R_\tau p$, (4) reconstruction uses the es-

---

**Algorithm 2** Residuals-to-Marginals (ReM)

---

**Input:** Marginal workload $\mathcal{W}$, arbitrary $\mathcal{S}$, measurements $z_{\tau,i} = R_\tau p + \xi_{\tau,i}$ for $\tau \in \mathcal{S}$, $i = 1, \ldots, k_\tau$, where $\xi_{\tau,i}$ comes from any noise distribution

1: Estimate $\hat{\alpha}_\tau \approx R_\tau p$ for $\tau \in \mathcal{S}$ by minimizing loss function $L_\tau(\alpha_\tau)$

2: Reconstruct $\hat{\mu}_\gamma = \sum_{\tau \in \mathcal{S}: \tau \subseteq \gamma} A_{\gamma,\tau} \hat{\alpha}_\tau$ for $\gamma \in \mathcal{W}$

---

timated residuals $\hat{\alpha}_\tau$ instead of the noisy measurements $z_\tau$. The loss function $L_\tau(\alpha_\tau)$ in Line 2 captures how well $\alpha_\tau$ explains the entire set of noisy measurements $\{z_{\tau,i}\}_{i=1,\ldots,k_\tau}$. For example, a typical choice is $L_\tau(\alpha_\tau) = -\sum_{i=1}^{k_\tau} \log p(z_{\tau,i}|R_\tau p = \alpha_\tau)$, the negative log-likelihood of the measurements.

The following result shows that solving the optimization problems in Line 1 is equivalent to finding a compact representation of a data distribution $\hat{p}$ that minimizes a global reconstruction loss and then using $\hat{p}$ to answer each marginal query.

**Theorem 1.** *Suppose $\hat{\alpha}_\tau$ minimizes $L_\tau(\alpha_\tau)$ over $\mathbb{R}^{m_\tau}$ for each $\tau \in \mathcal{S}$ and let $\hat{\alpha} = (\hat{\alpha}_\tau)_{\tau \in \mathcal{S}}$. Then Alg. 2 outputs $\hat{\mu}_\gamma = M_\gamma \hat{p}$, where $\hat{p} = R_{\mathcal{S}}^+ \hat{\alpha}$ is a global minimizer of the combined loss function $\sum_{\tau \in \mathcal{S}} L_\tau(R_\tau p)$ over $\mathbb{R}^n$.*

This result is proved (in Appendix D) by showing that $R_\tau \hat{p} = \hat{\alpha}_\tau$ for all $\tau$, and thus $\hat{p}$ optimizes each individual loss function $L_\tau$, and so must be a global minimizer. Proposition 1 then shows that $\hat{\mu}_\gamma = M_\gamma \hat{p} = M_\gamma R_{\mathcal{S}}^+ \hat{\alpha}$ has the form given in Line 2 of the algorithm.

## 4    Applications of ReM under Gaussian Noise

In this section, we apply ReM to reconstruct answers to marginal queries in various settings: (1) we reconstruct from residuals measured with Gaussian noise, (2) we reconstruct from marginals measured with isotropic Gaussian noise, and (3) we reconstruct non-negative answers from residuals measured with Gaussian noise.

### 4.1    Reconstruction under Gaussian noise

An instance of ReM that allows for efficient computation is when residuals are measured with Gaussian noise i.e., $z_{\tau,i} = R_\tau p + \xi_{\tau,i}$ where $\xi_{\tau,i} \sim \mathcal{N}(0, \Sigma_{\tau,i})$ and the loss function $L_\tau(\alpha_\tau)$ is the negative log-likelihood of the measurements. In this case, $\hat{\alpha} = (\hat{\alpha}_\tau)_{\tau \in \mathcal{S}}$ is the maximum likelihood estimate of the residual answers $\alpha = (\alpha_\tau)_{\tau \in \mathcal{S}}$. We refer to this setting as GReM-MLE (Gaussian ReM

---

**Algorithm 3** Gaussian ReM with Maximum Likelihood Estimation (GReM-MLE)

---

**Input:** Marginal workload $\mathcal{W}$, arbitrary $\mathcal{S}$, measurements $z_{\tau,i} = R_\tau p + \mathcal{N}(0, \Sigma_{\tau,i})$ for $\tau \in \mathcal{S}$, $i = 1, \ldots, k_\tau$

1: Estimate $\hat{\alpha}_\tau = \left(\sum_i \Sigma_{\tau,i}^{-1}\right)^{-1} \sum_i \Sigma_{\tau,i}^{-1} z_{\tau,i}$ for $\tau \in \mathcal{S}$

2: Reconstruct $\hat{\mu}_\gamma = \sum_{\tau \in \mathcal{S}: \tau \subseteq \gamma} A_{\gamma,\tau} \hat{\alpha}_\tau$ for $\gamma \in \mathcal{W}$

---

with Maximum Likelihood Estimation), shown in Alg. 3.

The loss function $L_\tau(\alpha_\tau)$ is a sum of quadratic forms given by $L_\tau(\alpha_\tau) = \sum_{i=1}^{k_\tau}(\alpha_\tau - z_{\tau,i})^\top \Sigma_{\tau,i}^{-1}(\alpha_\tau - z_{\tau,i})$. In this setting, the optimization problems in Line 1 of Alg. 2 have the closed-form solution $\hat{\alpha}_\tau = \left(\sum_i \Sigma_{\tau,i}^{-1}\right)^{-1}\sum_i \Sigma_{\tau,i}^{-1} z_{\tau,i}$, which is a form of inverse-variance weighting and can be verified by setting the gradient of the loss function to zero.

GReM-MLE improves computational tractability by reducing Alg. 2 to operations on matrices. Moreover, if the covariances among measurements of residual $R_\tau$ differ only by a constant for $\tau \in \mathcal{S}$, i.e., $\Sigma_{\tau,i} = \sigma_{\tau,i}^2 K_\tau$ where $\sigma_{\tau,i} \in \mathbb{R}$, then $\hat{\alpha}_\tau$ can be computed as a weighted average given by $\hat{\alpha}_\tau = (\sum_i \sigma_{\tau,i}^{-2})^{-1}\sum_i \sigma_{\tau,i}^{-2} z_{\tau,i}$. All instances of GReM-MLE considered throughout the paper satisfy this assumption of proportional covariances for each $\tau \in \mathcal{S}$.

## 4.2 Reconstruction from marginals

A common practice in existing mechanisms is to measure marginal queries with isotropic Gaussian noise [4, 7, 9, 11, 18, 20]. In this special case, the measurements can be converted to an equivalent set of residual measurements with independent Gaussian noise, allowing us to apply GReM-MLE.

The key observation is that a marginal query answer $\mu_\gamma = M_\gamma p$ for attribute set $\gamma$ can be used to derive residual answers $\alpha_\tau = R_\tau p$ for each $\tau \subseteq \gamma$ via the following Lemma (proved in Appendix D):

**Lemma 1.** *For $\tau \subseteq \gamma$, the residual $R_\tau$ can be recovered from the marginal $M_\gamma$ as*

$$R_\tau = A_{\gamma,\tau}^+ M_\gamma \text{ where } A_{\gamma,\tau}^+ = \bigotimes_{k=1}^{d} \begin{cases} D_{(k)} & k \in \tau \\ 1_k^T & k \in \gamma \setminus \tau \\ 1 & k \notin \gamma \end{cases}.$$

Whereas $A_{\gamma,\tau}$ maps answers from residual $R_\tau$ to answers to marginal $M_\gamma$, the matrix $A_{\gamma,\tau}^+$ maps answers from marginal $M_\gamma$ to residual $R_\tau$. Furthermore, $\mu_\gamma$ can be reconstructed from the set of all residuals $(\alpha_\tau)_{\tau \subseteq \gamma}$, so these residuals carry equivalent information to the marginal. Additionally, when the marginal is observed with isotropic noise as $y_\gamma = M_\gamma p + \mathcal{N}(0, \sigma_\gamma^2 I)$, the corresponding noisy residuals $A_{\gamma,\tau}^+ z_\tau$ are independent. As a consequence, we can decompose a noisy marginal measurement into a set of equivalent and independent noisy residual measurements.

**Theorem 2.** *Let $y_\gamma \sim \mathcal{N}(M_\gamma p, \sigma^2 I)$ be a noisy marginal measurement with isotropic Gaussian noise and let $z_\tau = A_{\gamma,\tau}^+ y_\gamma$ for each $\tau \subseteq \gamma$. Then noisy residual $z_\tau$ has distribution $\mathcal{N}(R_\tau p, \sigma^2 D_\tau D_\tau^\top \prod_{k \in \gamma \setminus \tau} n_k)$ and $z_\tau$ is independent of $z_{\tau'}$ for $\tau \neq \tau'$.*

*Furthermore, let $H_\gamma = (A_{\gamma,\tau}^+)_{\tau \subseteq \gamma}$ be the matrix mapping from $y_\gamma$ to $(z_\tau)_{\tau \subseteq \gamma}$. This matrix is invertible, which implies that*

$$\log \mathcal{N}(y_\gamma | M_\gamma p, \sigma^2 I) = \sum_{\tau \subseteq \gamma} \log \mathcal{N}\left(z_\tau \,\middle|\, R_\tau p, \sigma^2 D_\tau D_\tau^\top \prod_{k \in \gamma \setminus \tau} n_k\right) + \log|\det H_\gamma|. \quad (1)$$

Given a collection of noisy marginal measurements, we can apply the above decomposition to obtain a set of independent noisy residuals with proportional covariances. To reconstruct marginal answers, we can apply GReM-MLE to the residuals. Alg. 4 shows this decomposition and reconstruction. Equation (1) shows that the noisy residual measurements and noisy marginal measurements are equivalent from the perspective of finding the best data vector $p$ by maximum likelihood, because the log-likelihood of the residual measurements differs from the log-likelihood of the marginal measurement by a constant $\log|\det H_\gamma|$ that is independent of $p$, and measurements of marginals are each independent. A maximum likelihood estimate of $p$ from the marginal measurements $y$ is given by using the pseudoinverse of the measured workload to map noisy marginal measurements to a data vector. The following result shows that the method in Alg. 4 is equivalent to answering queries from this maximum likelihood estimate of the data vector given the marginal measurements when the marginals are measured with the same noise scale.

---

**Algorithm 4** Efficient Marginal Pseudoinversion (EMP)

---

**Input:** Marginal workload $\mathcal{W}$, measured marginals multiset $\mathcal{Q}$, measurements $y = (y_\gamma)_{\gamma \in \mathcal{Q}}$ where $y_\gamma = M_\gamma p + \mathcal{N}(0, \sigma^2 I)$ for $\gamma \in \mathcal{Q}$

**Output:** Marginal answers $(M_\gamma M_{\mathcal{Q}}^+ y)_{\gamma \in \mathcal{W}}$

  1: Initialize $\mathcal{S} = \emptyset$ and $k_\tau = 0$ for all $\tau$      ▷ Track measured residuals, lazy data structure for $k_\tau$

  2: **for** $\gamma \in \mathcal{Q}$ **do**

  3:     **for** $\tau \subseteq \gamma$ **do**

  4:         $\mathcal{S} = \mathcal{S} \cup \{\tau\}, \;\; k_\tau \leftarrow k_\tau + 1$

  5:         $z_{\tau, k_\tau} = A_{\gamma, \tau}^+ y_\gamma$                    ▷ Extract residual measurement from $y_\gamma$

  6:         $\sigma_{\tau, k_\tau}^2 = \sigma^2 \prod_{k \in \gamma \setminus \tau} n_k$                  ▷ Compute noise scale

  7:         $\Sigma_{\tau, k_\tau} = \sigma_{\tau, k_\tau}^2 D_\tau D_\tau^\top$               ▷ Proportional covariance

     **return** GReM-MLE$(\mathcal{W}, \mathcal{S}, z)$ where $z = (z_{\tau, i} : \tau \in \mathcal{S}, i = 1, \ldots, k_\tau)$

---

**Theorem 3** (Efficient pseudoinversion of marginal query matrix). *Let $M_{\mathcal{Q}} = (M_\gamma)_{\gamma \in \mathcal{Q}}$ be the query matrix for a multiset $\mathcal{Q}$ of marginals and let $y = (y_\gamma)_{\gamma \in \mathcal{Q}}$ be corresponding noisy marginal measurements with $y_\gamma = M_\gamma p + \mathcal{N}(0, \sigma^2 \mathcal{I})$. Let $\mathcal{S} = \{\tau \subseteq \gamma : \gamma \in \mathcal{Q}\}$ and for each $\tau \in \mathcal{S}$ let $\gamma_{\tau, i}$ be the ith marginal in $\mathcal{Q}$ containing $\tau$. Let $z_{\tau, i} = A_{\gamma_{\tau, i}, \tau}^+ y_{\gamma_{\tau, i}}$ be the residual measurement obtained from $\gamma_{\tau, i}$ and let $\Sigma_{\tau, i} = \sigma_{\tau, i}^2 D_\tau D_\tau^\top$ be its covariance where $\sigma_{\tau, i}^2 = \sigma^2 \prod_{k \in \gamma_{\tau, i} \setminus \tau} n_k$. Then, given any workload of marginal queries $\mathcal{W}$, for each $\gamma \in \mathcal{W}$, the marginal reconstruction $\hat{\mu}_\gamma$ obtained from Algorithm 3 on these residual measurements is equal to $M_\gamma M_{\mathcal{Q}}^+ y$.*

This result can be generalized to allow for differing noise scales between marginal measurements. We prove this result and discuss the generalized form of Theorem 3 in Appendix D.

### 4.3 Reconstruction with local non-negativity

It is often possible to improve accuracy of a differentially private mechanism by forcing its outputs to satisfy known constraints [4, 10, 21]. For our problem, true marginals are non-negative, so it is desirable to enforce non-negativity in their private estimates. To enforce non-negativity, instead of solving the separate problems in Line 1 of Alg. 2, we solve the following combined problem over the full vector $\alpha = (\alpha_\tau)_{\tau \in \mathcal{W}^\downarrow}$ of residuals:

$$\min_\alpha \sum_{\tau \in \mathcal{S}} L_\tau(\alpha_\tau) \quad \text{s.t.} \sum_{\tau \subseteq \gamma} A_{\gamma, \tau} \alpha_\tau \geq 0, \quad \forall \gamma \in \mathcal{W}. \tag{2}$$

Reconstruction of marginals then proceeds as in Line 2 of Alg. 2. The constraints in Eq. (2) ensure that the reconstructed marginals will be non-negative. We refer to this as *local non-negativity*, since this problem solves for a data distribution $\hat{p}$ that is non-negative for marginals in $\mathcal{W}$ rather than a data distribution with non-negative entries.

A natural setting to apply local non-negativity to ReM is under Gaussian noise with covariance $\Sigma_{\tau, i} = \sigma_{\tau, i}^2 D_\tau D_\tau^\top$ and $\sigma_{\tau, i}^2 \in \mathbb{R}$. Recall that marginals measured with isotropic Gaussian noise decompose into residuals with the above covariance structure. Our proposed application of local non-negativity in the Gaussian noise setting GReM-LNN (Gaussian ReM with local non-negativity) solves Eq. (2) for $L_\tau(\alpha_\tau) = \sum_{i=1}^{k_\tau} (\alpha_\tau - z_{\tau, i})^\top K_{\tau, i}^{-1} (\alpha_\tau - z_{\tau, i})$ and $K_{\tau, i} = 2^{|\tau|} D_\tau D_\tau^\top$. In the GReM-LNN setting, Eq. (2) is an convex program with linear constraints. Our implementation solves this problem using a scalable dual ascent algorithm (described in Appendix F) but could be solved in principle using standard optimizers, given sufficient resources [22]. With respect to the loss function $L_\tau(\alpha_\tau)$, adopting $2^{|\tau|}$ rather than Gaussian noise scale $\sigma_{\tau, i}^2$ is a heuristic that weights lower degree residual queries such as the total query and 1-way residuals more heavily than higher degree queries such as 3-way residuals. In contrast, using the Gaussian noise scale $\sigma_{\tau, i}^2$ obtained from both ResidualPlanner and the marginal decomposition in Theorem 2 weights higher degree residual

queries more than lower degree residuals. When enforcing local non-negativity, it is beneficial for reducing reconstruction error to allocate more weight to residuals that affect more marginals through reconstruction. The present choice of weights $2^{|\tau|}$ for GReM-LNN, however, remain a heuristic. We discuss this point further in Section 6.

### 4.4 Computational Complexity

We summarize the complexity results in Table 1. Formal statements and proofs appear in Appendix E. Let $\mathcal{S}$ be the set of measured residuals. To understand the results, suppose that $R_\tau$ is measured once, given by $z_\tau$, for each $\tau \in \mathcal{S}$. Recall from Proposition 1 that computing $A_{\gamma,\tau} z_\tau$ takes $\mathcal{O}(|\gamma| n_\gamma)$ time. This operation maps from the space of $\tau$-residuals to $\gamma$-marginals. To reconstruct an answer to the marginal query $M_\gamma$, we apply the invertible transformation from residuals to marginals by summing over contributions for each $\tau \subseteq \gamma$ to yield $\hat{\mu}_\gamma = \sum_{\tau \in \mathcal{S}:\tau \subseteq \gamma} A_{\gamma,\tau} z_\tau$. In the worst case, this requires computing $A_{\gamma,\tau} z_\tau$ for $2^{|\gamma|}$ residuals. Then the running time of reconstructing an answer to marginal $M_\gamma$ is $\mathcal{O}(|\gamma| n_\gamma 2^{|\gamma|})$. If $\mathcal{W}$ is a workload of marginals, then reconstructing answers to each $\gamma \in \mathcal{W}$ is $\mathcal{O}(\sum_{\gamma \in \mathcal{W}} |\gamma| n_\gamma 2^{|\gamma|})$. The following result shows that the complexity of reconstructing an answer to marginal $M_\gamma$ is almost linear with respect to domain size.

**Theorem 4.** *For $\varepsilon > 0$, reconstructing an answer to $M_\gamma$ is $o(n_\gamma^{1+\varepsilon})$ as $n_i \to \infty$ for some $i \in \gamma$.*

| Method | Running Time |
|---|---|
| GReM-MLE$(\mathcal{W}, \mathcal{S}, z)$ | $\mathcal{O}(\sum_{\gamma \in \mathcal{W}} |\gamma| n_\gamma 2^{|\gamma|})$ |
| EMP$(\mathcal{W}, \mathcal{Q}, y)$ | $\mathcal{O}(\sum_{\gamma \in \mathcal{W}} |\gamma| n_\gamma 2^{|\gamma|})$ |
| One Round of GReM-LNN$(\mathcal{W}, \mathcal{S}, z)$ | $\mathcal{O}(\sum_{\gamma \in \mathcal{W}} |\gamma| n_\gamma 2^{|\gamma|})$ |

Table 1: Summary of Complexity Results

GReM-MLE, given in Alg. 3, consists of two steps: estimating residual answers $\hat{\alpha}_\tau$ from residuals answers $z_{\tau,i}$ for $i = 1, \ldots, k_\tau$ and each $\tau \in S$, and reconstructing answers to marginal workload $\mathcal{W}$. Recall that we suppose that covariance is proportional among measurements of a given residual $R_\tau$, so $\hat{\alpha}_\tau$ can be computed in closed-form as a weighted average in $\mathcal{O}(n_\tau)$ time. Then GReM-MLE takes $\mathcal{O}(\sum_{\gamma \in \mathcal{W}} |\gamma| n_\gamma 2^{|\gamma|})$ time. The efficient marginal pseudoinversion, given in Alg. 4, first decomposes marginals and then applies GReM-MLE. Computing $A_{\gamma,\tau}^+ y_\gamma$ takes $\mathcal{O}(|\gamma| n_\gamma)$ time, so the running time of decomposing the marginal measurements is $\mathcal{O}(\sum_{\gamma \in \mathcal{Q}} |\gamma| n_\gamma 2^{|\gamma|})$ where $\mathcal{Q}$ is the set of marginals measured with isotropic Gaussian noise. Then the efficient marginal pseudoinversion is $\mathcal{O}(\sum_{\gamma \in \mathcal{W}} |\gamma| n_\gamma 2^{|\gamma|})$. Additionally, one round of GReM-LNN, given in Alg. 6, is $\mathcal{O}(\sum_{\gamma \in \mathcal{W}} |\gamma| n_\gamma 2^{|\gamma|})$.

## 5  Experiments

In this section, we measure the utility of GReM-MLE and GReM-LNN by incorporating them as a post-processing step into two mechanisms for privately answering marginals: (1) ResidualPlanner [6], and (2) a data-dependent mechanism we call Scalable MWEM. Both mechanisms measure queries with Gaussian noise and reconstruct answers to all three-way marginals for the given data domain. For the ResidualPlanner experiment, we measure residuals for all subsets of three or fewer attributes with Gaussian noise scales determined by ResidualPlanner. For the Scalable MWEM experiment, we measure the total query and a subset of the 3-way marginals in the data domain with isotropic Gaussian noise and reconstruct answers to all 3-ways marginals using the efficient marginal pseudoinversion in Alg. 4. We fully describe Scalable MWEM in Appendix G.

We compare average $\ell_1$ error with respect to the reconstructed marginals of the base mechanism to post-processing with GReM-LNN and two heuristics that enforce non-negativity by truncating negative values to zero (Trunc) and truncating to zero then rescaling (Trunc+Rescale). For the Scalable MWEM experiment, we additionally compare to a well-studied reconstruction mechanism Private-PGM [12]. We run these methods on four datasets of varying size and scale, Titanic [23], Adult [24],

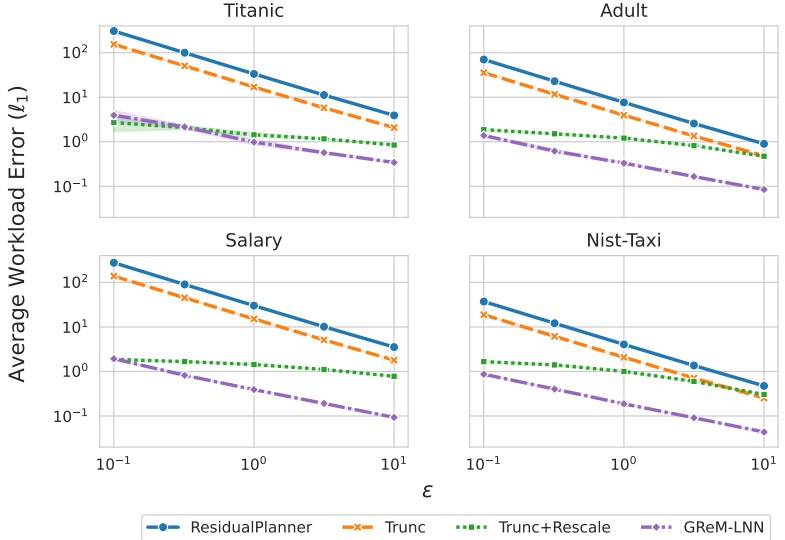

Figure 2: Average $\ell_1$ workload error on all 3-way marginals across five trials and privacy budgets $\epsilon \in \{0.1, 0.31, 1, 3.16, 10\}$ and $\delta = 1 \times 10^{-9}$ for ResidualPlanner.

Salary [25], and Nist-Taxi [26], and various practical privacy regimes, $\epsilon \in \{0.1, 0.31, 1, 3.16, 10\}$ and $\delta = 1 \times 10^{-9}$. For each setting, we run five trials and report the average error of each method as well as minimum/maximum bands. Additional details are provided in Appendix H.

## 5.1 ResidualPlanner Results

Fig. 2 displays results for the ResidualPlanner experiment. Across all privacy budgets and datasets considered, GReM-LNN significantly reduces workload error on the reconstructed marginals compared to ResidualPlanner. Averaging over all settings and trials, GReM-LNN reduces ResidualPlanner workload error by a factor of $44.0\times$. With respect to the heuristic methods, GReM-LNN reconstructs marginals with lower error than Trunc across all privacy budgets and datasets. Except at the highest privacy regime considered ($\epsilon = 0.1$) on Titanic and Salary, GReM-LNN yields lower error than Trunc+Rescale. Averaging over all settings and trials, GReM-LNN has lower workload error by a factor of $17.6\times$ compared to Trunc and $3.2\times$ compared to Trunc+Rescale. Note that GReM-MLE is omitted from Fig. 2 since ResidualPlanner is the maximum likelihood reconstruction for its measurements. Appendix I reports results for this experiment with respect to $\ell_2$ workload error, which are consistent with the present findings.

## 5.2 Scalable MWEM Results

Fig. 3 displays results for the Scalable MWEM experiment for 30 rounds of measurements. Observe that Scalable MWEM runs for the settings considered, which would be infeasible for the original MWEM mechanism due to large data domains. Of all methods considered, Private-PGM yields the greatest reduction in workload error in settings where it ran; however, Private-PGM failed due to exceeding memory resources (20 GB) at 30 rounds on Adult, Salary, and Nist-Taxi in all trials. In Appendix I, we report the settings in which Private-PGM successfully ran across 10, 20, and 30 rounds of Scalable MWEM.

With respect to GReM-LNN, the findings from the prior experiment agree with the present results. Across all privacy budgets and datasets considered, GReM-LNN significantly reduces workload error on the reconstructed marginals compared to Scalable MWEM. Averaging over all settings and trials, GReM-LNN reduces Scalable MWEM workload error by a factor of $12.3\times$. Averaging over all settings and trials, GReM-LNN has lower workload error by a factor of $1.1\times$ compared to Trunc+Rescale. Note that we suppress results for Trunc due to space. Appendix I reports results for this experiment with respect to $\ell_2$ workload error, which are consistent with the present findings.

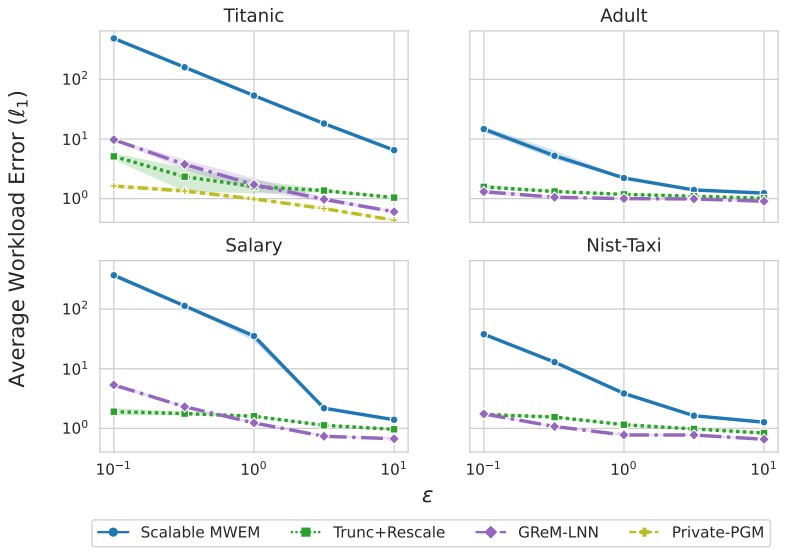

Figure 3: Average $\ell_1$ workload error on all 3-way marginals across five trials and privacy budgets $\epsilon \in \{0.1, 0.31, 1, 3.16, 10\}$ and $\delta = 1 \times 10^{-9}$ for Scalable MWEM with 30 rounds of measurements.

## 6  Discussion

We develop ReM, a method for reconstructing answers to marginal queries that scales to large data domains. We also introduce a tractable method to incorporate local non-negativity that significantly improves reconstruction quality. Finally, we show that ReM can be used to improve the existing query answering mechanisms ResidualPlanner and a scalable version of MWEM.

**Limitations.** Many data-dependent query answering mechanisms also generate synthetic data. In some cases, practitioners utilize these mechanisms primarily in order to use the synthetic data for downstream tasks such as training a machine learning model [27, 28]. For those users, the fact that ReM does not generate synthetic data would be an important limitation. A broader limitation, which is common to many methods in this field, is lack of support for continuous data. Marginal and residual queries are only defined on discrete domains so continuous attributes need to be discretized.

**Future Work and Broader Impacts.** While developing effective algorithms for privacy-preserving data analysis is generally beneficial, it is known that these methods can lead to unfair outcomes [29]. One direction for future work is to further understand the fairness properties of the methods we present and how to mitigate any undesirable outcomes. Another direction for future work is further understanding the weighting scheme used in GReM-LNN to apply local non-negativity. Preliminary experiments show that weighting lower-order residual queries more highly in the loss function yields reconstructed answers with lower workload error as well as faster and more reliable convergence of the optimization routine. In general, the relationship between residual weights in the loss function, optimizer convergence, and reconstruction quality is not well understood.

## Acknowledgments and Disclosure of Funding

This work was supported by the National Science Foundation under grants CNS-1931686 and CNS-2317232 (Kifer); CCF-2046235 (Musco); and IIS-1749854 and DBI-2210979 (Sheldon).

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

# A  Kronecker Products

Kronecker products are a convenient way to represent highly structured matrices. Let $A$ be an

$m_a \times n_a$ matrix $A = \begin{bmatrix} a_{1,1} & \cdots & a_{1,n_a} \\ \vdots & & \vdots \\ a_{m_a,1} & \cdots & a_{m_a,n_a} \end{bmatrix}$ and $B$ be a $m_b \times n_b$ matrix. Then the Kronecker

product of $A$ with $B$ is an $m_a m_b \times n_a n_b$ matrix given by $A \otimes B = \begin{bmatrix} a_{1,1}B & \cdots & a_{1,n_a}B \\ \vdots & & \vdots \\ a_{m_a,1}B & \cdots & a_{m_a,n_a}B \end{bmatrix}$.

Kronecker products provide a compact representation of matrices by representing exponentially-many entries of $A \otimes B$ with linearly-many entries in $A$ and $B$. For the Kronecker product of a sequence of matrices $A_1, \ldots, A_d$, we use the notation

$$\bigotimes_{i=1}^{d} A_i = A_1 \otimes \cdots \otimes A_d$$

The Kronecker product is associative, so pairwise products can be taken in any order.

Kronecker products additionally possess useful algebraic properties. Let $(\cdot)^+$ denote Moore-Penrose pseudoinverse.

**Proposition 3.** *(Kronecker Product Properties) Let $A = \bigotimes_{i=1}^{d} A_i$ and $B = \bigotimes_{j=1}^{d} B_j$. Then the following properties hold:*

*1. $A^\top = \bigotimes_{i=1}^{d} A_i^\top$.*

*2. $A^+ = \bigotimes_{i=1}^{d} A_i^+$.*

*3. If $A_i$ and $B_i$ are compatible for multiplication for $i = 1, \ldots, d$, then $AB = \bigotimes_{i=1}^{d} A_i B_i$.*

There are efficient algorithms for matrix-vector multiplication utilizing Kronecker structure such as Alg. 5. Let $A = \bigotimes_{i=1}^{\ell} A_i$ be a Kronecker structured matrix where $A_i$ is a matrix of size $a_i \times b_i$ so that $A$ has size $a \times b$ with $a = \prod_{i=1}^{\ell} a_i$ and $b = \prod_{i=1}^{\ell} b_i$.

---

**Algorithm 5** Kronecker Matrix-Vector Product [20, 30]

---

**Input:** Matrix $A = \bigotimes_{i=1}^{\ell} A_i$, vector $x$
    $a_i, b_i = \text{SHAPE}(A_i)$
    $r = \prod_{i=1}^{\ell} b_i$
    $x_1 = x$
    **for** $i = 1, \ldots, \ell$ **do**
        $Z = \text{RESHAPE}(x_i, b_i, r/b_i)$
        $r = r \cdot a_i/b_i$
        $x_{i+1} = \text{RESHAPE}(A_i Z, r, 1)$
    **return** $x_{\ell+1}$

---

# B  Differential Privacy

Let us begin by introducing a useful variant of differential privacy: zero-concentrated differential privacy (zCDP).

**Definition 2.** (Zero-Concentrated Differential Privacy; [31]) Let $\mathcal{M} : \mathcal{X} \to \mathcal{Y}$ be a randomized mechanism. For any neighboring datasets $p, p'$ that differ by at most one record, denoted $p \sim p'$, and all measurable subsets $S \subseteq \mathcal{Y}$: if $D_\gamma(\mathcal{M}(p)||\mathcal{M}(p')) \leq \rho\gamma$ for all $\gamma \in (1, \infty)$ where $D_\gamma$ is the $\gamma$-Renyi divergence between distributions $\mathcal{M}(p), \mathcal{M}(p')$, then $\mathcal{M}$ satisfies $\rho$-zCDP.

While $(\epsilon, \delta)$-DP is a more common notion, it is often more convenient to work with zCDP. There exists a conversion from zCDP to $(\epsilon, \delta)$-DP.

**Proposition 4** (zCDP to DP Conversion; [32]). *If mechanism $\mathcal{M}$ satisfies $\rho$-zCDP, then $\mathcal{M}$ satisfies $(\epsilon, \delta)$-DP for any $\epsilon > 0$ and $\delta = \min_{\alpha > 1} \frac{\exp((\alpha-1)(\alpha\rho - \epsilon))}{\alpha - 1} \left(1 - \frac{1}{\alpha}\right)^{\alpha}$.*

Next, we introduce two building block mechanisms. An important quantity in analyzing the privacy of a mechanism is sensitivity. The $\ell_k$ sensitivity of a function $f : \mathcal{X} \to \mathbb{R}$ is given by $\Delta_k(f) = \max_{p \sim p'} \|f(p) - f(p')\|_k$. If $f$ is clear from the context, we write $\Delta_k$.

**Proposition 5** (zCDP of Gaussian mechanism; [31]). *Let $W$ be an $m \times n$ workload. Given data vector $p$, the Gaussian mechanism adds i.i.d. Gaussian noise to $Wp$ with scale parameter $\sigma$ i.e., $\mathcal{M}(p) = Wp + \sigma\Delta_2(W)\mathcal{N}(0, \mathbb{I})$, where $\mathbb{I}$ is the $m \times m$ identity matrix. Then the Gaussian Mechanism satisfies $\frac{1}{2\sigma^2}$-zCDP.*

**Proposition 6** (zCDP of correlated Gaussian mechanism; [33]). *Let $W$ be an $m \times n$ workload. Given data vector $p$, the correlated Gaussian mechanism adds Gaussian noise to $Wp$ with covariance matrix $\Sigma$ i.e., $\mathcal{M}(p) = Mp + \mathcal{N}(0, \Sigma)$. The correlated Gaussian mechanism satisfies $\frac{\gamma}{2}$-zCDP where $\gamma$ is the largest diagonal element of $M^\top \Sigma^{-1} M$.*

**Proposition 7** (zCDP of exponential mechanism; [34, 35]). *Let $\epsilon > 0$ and $\text{Score}_r : \mathcal{X} \to \mathbb{R}$ be a quality score of candidate $r \in \mathcal{R}$ for data vector $p$. Then the exponential mechanism outputs a candidate $r \in \mathcal{R}$ according to the following distribution: $\Pr(\mathcal{M}(p) = r) \propto \exp\left(\frac{\epsilon}{2\Delta_1}\text{Score}_r(p)\right)$. The exponential mechanism satisfies $\frac{\epsilon^2}{8}$-zCDP.*

Adaptive composition and post-processing are two important properties of differential privacy that allow us to construct complex mechanisms from the above building blocks. Let us state these results for zCDP.

**Proposition 8** (zCDP Properties; [31, 35]). *zCDP satisfies these two properties of differential privacy:*

1. *(Adaptive Composition) Let $\mathcal{M}_1 : \mathcal{X} \to \mathcal{Y}_1$ satisfy $\rho_1$-zCDP and $\mathcal{M}_2 : \mathcal{X} \times \mathcal{Y}_1 \to \mathcal{Y}_2$ satisfy $\rho_2$-zCDP. The mechanism $p \mapsto \mathcal{M}_2(p, \mathcal{M}_1(p))$ satisfies $(\rho_1 + \rho_2)$-zCDP.*

2. *(Post-processing) Let $\mathcal{M}_1 : \mathcal{X} \to \mathcal{Y}$ satisfy $\rho$-zCDP and $f : \mathcal{Y} \to \mathcal{Z}$ be a randomized algorithm. Then $\mathcal{M} : \mathcal{X} \to \mathcal{Z} = f \circ \mathcal{M}_1$ satisfies $\rho$-zCDP.*

## C   Relationship between Marginals and Residuals

In this section, we prove Proposition 1, which provides a relationship between marginals and residuals. Before proving this result, let us consider residual workloads as well as the subtraction matrix $D_{(k)}$.

Let us state some properties of residuals.

**Proposition 9** (Residual Properties; [6, 13, 14]). *Let $\Omega$ be the set of all tuples of attributes for a given data universe $\mathcal{X}$.*

1. *$R_\tau$ is an $m_\tau \times n$ matrix with full row rank.*

2. *$R_\tau, R_{\tau'}$ are mutually orthogonal for $\tau \neq \tau'$ i.e. $R_\tau R_{\tau'}^\top = \mathbf{0}$.*

3. *$R_\tau, M_{\tau'}$ are mutually orthogonal for $\tau \not\subseteq \tau'$ i.e. $R_\tau M_{\tau'}^\top = \mathbf{0}$.*

4. *$(R_\tau)_{\tau \in \Omega}$ spans $\mathbb{R}^n$.*

**Lemma 2.** *Data vector $p \in \mathbb{R}^n$ can be decomposed uniquely as follows: $p = \sum_{\tau \in \Omega} R_\tau^\top v_\tau$ for $v_\tau \in \mathbb{R}^{m_\tau}$.*

*Proof.* Let $p_\tau = R_\tau^+ R_\tau p$ be the projection of $p$ onto the row-space of $R_\tau$. By Proposition 9, $p = \sum_{\tau \in \Omega} p_\tau$. Let $v_\tau \in \mathbb{R}^{m_\tau}$ be such that $p_\tau = R_\tau^\top v_\tau$. Since $R_\tau$ is full row rank, $v_\tau$ is unique. $\square$

Now, let us consider $D_{(k)}^+$. Recall that $D_{(k)}$ is an $n_k - 1 \times n_k$ matrix given by

$$D_{(k)} = \begin{bmatrix} 1 & -1 & 0 & \cdots & 0 \\ 0 & 1 & -1 & \cdots & 0 \\ \vdots & \vdots & \vdots & & \vdots \\ 0 & \cdots & \cdots & 1 & -1 \end{bmatrix}.$$

The pseudoinverse of $D_{(k)}$ is known in closed-form:

$$D_{(k)}^+ = \frac{1}{n_k} \begin{bmatrix} n_k - 1 & n_k - 2 & \cdots & 1 \\ -1 & n_k - 2 & \cdots & 1 \\ -1 & -2 & \cdots & 1 \\ \vdots & \vdots & & \vdots \\ -1 & -2 & \cdots & -(n_k - 1) \end{bmatrix}$$

$$= (1/n_k)(1_k u_k^\top - n_k C_k),$$

where $u_k = \begin{bmatrix} n_k - 1 \\ n_k - 2 \\ \vdots \\ 1 \end{bmatrix}$ and $C_k$ is the $n_k \times n_k - 1$ lower triangular matrix of ones.

Continuing the example from Fig. 1,

$$D_{(k)} = \begin{bmatrix} 1 & -1 & 0 \\ 0 & 1 & -1 \end{bmatrix} \qquad D_{(k)}^+ = \frac{1}{3} \begin{bmatrix} 2 & 1 \\ -1 & 1 \\ -1 & -2 \end{bmatrix}.$$

**Proposition 1.** *Let $R_S = (R_\tau)_{\tau \in S}$ be a combined workload of residual queries for all $\tau$ in a collection $S \subseteq 2^{[d]}$, where the individual matrices $R_\tau$ are stacked vertically. The size of $R_S$ is $m \times n$ where $m = \sum_{\tau \in S} m_\tau$. Then for any $z = (z_\tau)_{\tau \in S} \in \mathbb{R}^m$ and any $\gamma$, it holds that*

$$M_\gamma R_S^+ z = \sum_{\tau \in S, \tau \subseteq \gamma} A_{\gamma,\tau} z_\tau, \qquad where \ A_{\gamma,\tau} := \bigotimes_{k=1}^d \begin{cases} D_{(k)}^+ & k \in \tau \\ (1/n_k) 1_k & k \in \gamma \setminus \tau \\ 1 & k \notin \gamma \end{cases} \quad for \ \tau \subseteq \gamma.$$

*The matrix $A_{\gamma,\tau}$ has size $n_\gamma \times m_\tau$ and maps from the space of $\tau$-residuals to the space of $\gamma$-marginals. The running time to compute $A_{\gamma,\tau} z_\tau$ is $\mathcal{O}(|\gamma| n_\gamma)$.*

*Proof of Proposition 1.* First note that $R_S^+$ is the pseudoinverse of a block matrix. In general the pseudoinverse of a vertical block matrix involves the pseudoinverse of each block multiplied by a projection matrix [36]. In this case each block is a residual query, as discussed in Proposition 9, these query matrices are mutually orthogonal so the pseudoinverse $R_S^+$ has the form $(R_\tau^+)_{\tau \in S}^T$. Here, the combined query matrix $R_S$ is constructed by stacking the blocks $R_\tau$ vertically and the combined pseudoinverse $R_S^+$ stacks the blocks $R_\tau^+$ horizontally. Given this block structure of $R_S^+$ we can write

$$R_S^+ z = \sum_{\tau \in S} R_\tau^+ z_\tau \quad \implies \quad M_\gamma R_S^+ z = \sum_{\tau \in S} M_\gamma R_\tau^+ z_\tau. \tag{3}$$

Another relevant property of residual queries given in Proposition 9 is that $R_\tau M_{\tau'}^\top = \mathbf{0}$ for $\tau \not\subseteq \tau'$. When we drop these orthogonal queries from the summation, we get $M_\gamma R_S^+ z = \sum_{\tau \in S, \tau \subseteq \gamma} M_\gamma R_\tau^+ z_\tau$. When computing the product $M_\gamma R_\tau^+$, several properties of Kronecker products given in Proposition 3 are relevant. The first is that $(A \otimes B)^+ = A^+ \otimes B^+$. Applying this property gives

$$R_\tau^+ = \bigotimes_{k=1}^d \begin{cases} D_{(k)}^+ & k \in \tau \\ (1_k^\top)^+ & k \notin \tau \end{cases}. \tag{4}$$

The next property is that when when $A$ and $B$ both have compatible Kronecker structure, $AB = \bigotimes_i A_i B_i$. Both $M_\gamma$ and $R_\tau^+$ have compatible Kronecker structure so we can write

$$M_\gamma R_\tau^+ = \bigotimes_{k=1}^d \begin{cases} I_k D_{(k)}^+ & k \in \tau \\ I_k \left(1_k^\top\right)^+ & k \in \gamma \setminus \tau \\ 1_k^\top \left(1_k^\top\right)^+ & k \notin \gamma \end{cases}. \tag{5}$$

To evaluate this, notice that $(1_k^T)^+ = 1_k(1_k^T 1_k)^{-1} = 1_k/n_k$ and $1_k^\top(1_k/n_k) = 1$. Plugging this into the equation above we get

$$A_{\gamma,\tau} = M_\gamma R_\tau^+ = \bigotimes_{k=1}^{d} \begin{cases} D_{(k)}^+ & k \in \tau \\ 1_k/n_k & k \in \gamma \setminus \tau \\ 1 & k \notin \gamma \end{cases}. \tag{6}$$

Finally, this gives the full result that $M_\gamma R_\mathcal{S}^+ z = \sum_{\tau \in \mathcal{S}, \tau \subseteq \gamma} A_{\gamma,\tau} z_\tau$. $\qquad\square$

We prove the time complexity result for $A_{\gamma,\tau} z_\tau$ in Appendix E.

## D   ReM Proofs

In this section, we prove results related to ReM from Sections 3 and 4.

**Theorem 1.** *Suppose $\hat\alpha_\tau$ minimizes $L_\tau(\alpha_\tau)$ over $\mathbb{R}^{m_\tau}$ for each $\tau \in \mathcal{S}$ and let $\hat\alpha = (\hat\alpha_\tau)_{\tau \in \mathcal{S}}$. Then Alg. 2 outputs $\hat\mu_\gamma = M_\gamma \hat p$, where $\hat p = R_\mathcal{S}^+ \hat\alpha$ is a global minimizer of the combined loss function $\sum_{\tau \in \mathcal{S}} L_\tau(R_\tau p)$ over $\mathbb{R}^n$.*

*Proof.* Suppose $\hat\alpha_\tau$ minimizes $L_\tau$ for all $\tau$ and let $\hat p = R_\mathcal{S}^+ \hat\alpha$. Then for any $p \in \mathbb{R}^n$

$$\sum_{\tau \in \mathcal{S}} L_\tau(R_\tau \hat p) = \sum_{\tau \in \mathcal{S}} L_\tau(R_\tau R_\mathcal{S}^+ \hat\alpha) \overset{(\star)}{=} \sum_{\tau \in \mathcal{S}} L_\tau(\hat\alpha_\tau) \leq \sum_{\tau \in \mathcal{S}} L_\tau(R_\tau p). \tag{7}$$

We will justify Equality $(\star)$ below. The inequality holds because $\hat\alpha_\tau$ minimizes $L_\tau$. Thus, Equation (7) shows that $\hat p$ minimizes the combined loss $\sum_{\tau \in \mathcal{S}} L_\tau(R_\tau p)$.

To justify Equality $(\star)$, first observe that $R_\mathcal{S} R_\mathcal{S}^+ \hat\alpha = \hat\alpha$ because $R_\mathcal{S}$ has full row rank and thus $R_\mathcal{S} R_\mathcal{S}^+ = I$. We can see $R_\mathcal{S}$ has full row rank by Proposition 9: each block of rows corresponding to one residual has full row rank and these blocks are orthogonal. The equality $R_\tau R_\mathcal{S}^+ \hat\alpha = \hat\alpha_\tau$ is obtained by selecting the block of rows corresponding to residual $\tau$ from the equality $R_\mathcal{S} R_\mathcal{S}^+ \hat\alpha = \hat\alpha$. $\qquad\square$

**Lemma 1.** *For $\tau \subseteq \gamma$, the residual $R_\tau$ can be recovered from the marginal $M_\gamma$ as*

$$R_\tau = A_{\gamma,\tau}^+ M_\gamma \text{ where } A_{\gamma,\tau}^+ = \bigotimes_{k=1}^{d} \begin{cases} D_{(k)} & k \in \tau \\ 1_k^T & k \in \gamma \setminus \tau \\ 1 & k \notin \gamma \end{cases}.$$

*Proof of Lemma 1.* Recall that we defined $A_{\gamma,\tau} = M_\gamma R_\tau^+$. Then $A_{\gamma,\tau}^+ = R_\tau M_\gamma^+$. Observe the following:

$$A_{\gamma,\tau}^+ M_\gamma = \bigotimes_{k=1}^{d} \begin{cases} D_{(k)} I_k & k \in \tau \\ 1_k^\top I_k & k \in \gamma \setminus \tau \\ 1 \cdot 1_k^\top & k \notin \gamma \end{cases}$$

$$= \bigotimes_{k=1}^{d} \begin{cases} D_{(k)} & k \in \tau \\ 1_k^\top & k \notin \tau \end{cases}$$

$$= R_\tau$$

$\qquad\square$

**Theorem 2.** *Let $y_\gamma \sim \mathcal{N}(M_\gamma p, \sigma^2 I)$ be a noisy marginal measurement with isotropic Gaussian noise and let $z_\tau = A_{\gamma,\tau}^+ y_\gamma$ for each $\tau \subseteq \gamma$. Then noisy residual $z_\tau$ has distribution $\mathcal{N}(R_\tau p, \sigma^2 D_\tau D_\tau^\top \prod_{k \in \gamma \setminus \tau} n_k)$ and $z_\tau$ is independent of $z_{\tau'}$ for $\tau \neq \tau'$.*

*Furthermore, let $H_\gamma = (A^+_{\gamma,\tau})_{\tau \subseteq \gamma}$ be the matrix mapping from $y_\gamma$ to $(z_\tau)_{\tau \subseteq \gamma}$. This matrix is invertible, which implies that*

$$\log \mathcal{N}(y_\gamma | M_\gamma p, \sigma^2 I) = \sum_{\tau \subseteq \gamma} \log \mathcal{N}\left(z_\tau \,\middle|\, R_\tau p,\, \sigma^2 D_\tau D_\tau^\top \prod_{k \in \gamma \setminus \tau} n_k\right) + \log |\det H_\gamma|. \quad (1)$$

*Proof of Theorem 2.* Since $y_\tau \sim \mathcal{N}(M_\gamma p, \sigma^2 I)$ and $z_\tau = A^+_{\gamma,\tau} y_\tau$, standard properties of normal distributions give that $z_\tau \sim \mathcal{N}(A^+_{\gamma,\tau} M_\gamma p, \sigma^2 A^+_{\gamma,\tau}(A^+_{\gamma,\tau})^\top)$. By Lemma 1, the mean is equal to $R_\tau p$, as stated. For the covariance

$$A^+_{\gamma,\tau}(A^+_{\gamma,\tau})^\top = \begin{cases} D_{(k)} D_{(k)}^\top & k \in \tau \\ 1_k^\top 1_k & k \in \gamma \setminus \tau \\ 1 & k \notin \gamma \end{cases}$$

$$= \prod_{k \in \gamma \setminus \tau} n_k \cdot \bigotimes_{k=1}^d \begin{cases} D_{(k)} D_{(k)}^\top & k \in \tau \\ 1 & k \notin \tau \end{cases}$$

$$= D_\tau D_\tau^\top \prod_{k \in \gamma \setminus \tau} n_k$$

so the covariance is $\sigma^2 D_\tau D_\tau^\top \prod_{k \in \gamma \setminus \tau} n_k$, as stated.

For $\tau \neq \tau'$ the vectors $z_\tau$ and $z_{\tau'}$ are jointly normal with covariance $\sigma^2 A^+_{\gamma,\tau}(A^+_{\gamma,\tau'})^\top$. We will show that $A^+_{\gamma,\tau}(A^+_{\gamma,\tau'})^\top$ is a matrix of zeros, so the covariance matrix is identically zero and the vectors are independent. By the Kronecker structure,

$$A^+_{\gamma,\tau}(A^+_{\gamma,\tau'})^\top = \bigotimes_{k=1}^d \begin{cases} D_{(k)} D_{(k)}^\top & k \in \tau \cap \tau' \\ 1_k^\top D_{(k)}^\top & k \in \tau' \setminus \tau \\ D_{(k)} 1_k & k \in \tau \setminus \tau' \\ 1_k^\top 1_k & k \in \gamma \setminus (\tau \cup \tau') \\ 1 & k \notin \gamma \end{cases}$$

Observe that $D_{(k)} 1_k = 0$ is a vector of zeros because the rows of $D_{(k)}$ sum to zero, and similarly $1_k^\top D_{(k)}^\top$ is a row vector of zeros. Thus, any $k$ in the symmetric difference $(\tau' \setminus \tau) \cup (\tau \setminus \tau')$ will contribute an all zeros matrix to the Kronecker product and cause $A^+_{\gamma,\tau}(A^+_{\gamma,\tau'})^\top$ to be an all zeros matrix. But there must be at least one $k$ in the symmetric difference because $\tau \neq \tau'$. This proves that the covariance matrix is identically zero, as desired.

We will next show that the mapping $H_\gamma$ is invertible. $H_\gamma$ is a matrix with blocks $A^+_{\gamma,\tau}$ for each $\tau \subseteq \gamma$, stacked vertically, and $n_\gamma = \prod_{k \in \gamma} n_k$ columns. From the definition of the block $A^+_{\gamma,\tau}$, we can see it has $m_\tau = \prod_{k \in \tau}(n_k - 1)$ rows and is of full row rank because $D_{(k)}$ is a full rank matrix with $n_k - 1$ rows and the other matrices in the Kronecker product have only one row. We showed above that $A^+_{\gamma,\tau}(A^+_{\gamma,\tau'})^\top = 0$ for $\tau \neq \tau'$, which means that the blocks of $H_\gamma$ have mutually orthogonal rows, and combined with the fact that each block has full row rank this means that $H_\gamma$ has rank equal to the total number of rows. This number of rows is $\sum_{\tau \subseteq \gamma} m_\tau = \sum_{\tau \subseteq \gamma} \prod_{k \in \tau}(n_k - 1) = \prod_{k \in \gamma} n_k = n_\gamma$, which equals the number of columns, and therefore $H_\gamma$ invertible.[2]

Now, given what we have shown so far, we will write two different expressions for the log-probability density function $\log p_z(z)$ where $z = (z_\tau)_{\tau \subseteq \gamma}$. First, we have already derived the joint multivariate distribution of $z$, which, due to independence, has log-density

$$\log p_z(z) = \sum_{\tau \subseteq \gamma} \log \mathcal{N}\left(z_\tau \,\middle|\, R_\tau p,\, \sigma^2 D_\tau D_\tau^\top \prod_{k \in \gamma \setminus \tau} n_k\right).$$

---

[2] To see that $\sum_{\tau \subseteq \gamma} \prod_{k \in \tau}(n_k - 1) = \prod_{k \in \gamma} n_k$, observe that $n_\gamma = \prod_{k \in \gamma} n_k$ counts the number of ways to map each $k \in \gamma$ to a value $i \in \{1, \ldots, n_k\}$. Equivalently, we may consider selecting a subset $\tau \subseteq \gamma$, assigning each $k \in \tau$ to the value 1, and then assigning each $k \notin \tau$ to one of the remaining values in $\{2, \ldots, n_k\}$. The number of ways to do this is $\sum_{\tau \subseteq \gamma} \prod_{k \in \tau}(n_k - 1) = \prod_{k \in \gamma} n_k$.

Second, because $z = H_\gamma y_\gamma$ for the multivariate normal random variable $y_\gamma$, the change of variable formula for probability densities gives that

$$\log p_z(z) = \log \mathcal{N}(y_\gamma | M_\gamma p, \sigma^2 I) - \log |\det H_\gamma|.$$

Equating these two expressions gives Equation (1), which completes the proof. $\qquad\square$

**Theorem 3** (Efficient pseudoinversion of marginal query matrix)**.** *Let $M_\mathcal{Q} = (M_\gamma)_{\gamma \in \mathcal{Q}}$ be the query matrix for a multiset $\mathcal{Q}$ of marginals and let $y = (y_\gamma)_{\gamma \in \mathcal{Q}}$ be corresponding noisy marginal measurements with $y_\gamma = M_\gamma p + \mathcal{N}(0, \sigma^2 \mathcal{I})$. Let $\mathcal{S} = \{\tau \subseteq \gamma : \gamma \in \mathcal{Q}\}$ and for each $\tau \in \mathcal{S}$ let $\gamma_{\tau,i}$ be the $i$th marginal in $\mathcal{Q}$ containing $\tau$. Let $z_{\tau,i} = A^+_{\gamma_{\tau,i},\tau} y_{\gamma_{\tau,i}}$ be the residual measurement obtained from $\gamma_{\tau,i}$ and let $\Sigma_{\tau,i} = \sigma^2_{\tau,i} D_\tau D_\tau^\top$ be its covariance where $\sigma^2_{\tau,i} = \sigma^2 \prod_{k \in \gamma_{\tau,i} \setminus \tau} n_k$. Then, given any workload of marginal queries $\mathcal{W}$, for each $\gamma \in \mathcal{W}$, the marginal reconstruction $\hat\mu_\gamma$ obtained from Algorithm 3 on these residual measurements is equal to $M_\gamma M_\mathcal{Q}^+ y$.*

*Proof.* By standard properties of the pseudoinverse, $M_\mathcal{Q}^+ y$ is the unique vector that minimizes $\mathrm{SE}(p) = \|M_\mathcal{Q} p - y\|_2^2$ and is in the row span of $M_\mathcal{Q}$. We will show that the vector $R_\mathcal{S}^+ \hat\alpha$ satisfies both properties, where $\hat\alpha = (\hat\alpha_\tau)_{\tau \in \mathcal{S}}$ is constructed in Algorithm 3, and thus $R_\mathcal{S}^+ \hat\alpha = M_\mathcal{Q}^+ y$. Then, by Proposition 1, the reconstructed marginal $\hat\mu_\gamma$ in Algorithm 3 is equal to $M_\gamma R_\mathcal{S}^+ \hat\alpha$ and hence also equal to $M_\gamma M_\mathcal{Q}^+ y$, as claimed.

We will first show $R_\mathcal{S}^+ \hat\alpha$ minimizes $\mathrm{SE}(p)$. Observe that the $\mathrm{SE}(p)$ is equivalent to the negative log-likelihood $\mathcal{L}_y(p)$ of the marginal measurements $y$:

$$\begin{aligned}
\mathrm{SE}(p) &= \|M_\mathcal{Q} p - y\|_2^2 \\
&= \sum_{\gamma \in \mathcal{Q}} \|M_\gamma p - y_\gamma\|_2^2 \\
&= -2\sigma^2 \sum_{\gamma \in \mathcal{Q}} \log \mathcal{N}(y_\gamma | M_\gamma p, \sigma^2 I) + \mathrm{const.} \\
&= 2\sigma^2 \mathcal{L}_y(p) + \mathrm{const.}
\end{aligned}$$

Therefore $\mathrm{SE}(p)$ and $\mathcal{L}_y(p)$ have the same minimizers.

Then, by Theorem 2,

$$\begin{aligned}
\mathcal{L}_y(p) &= \sum_{\gamma \in \mathcal{Q}} -\log \mathcal{N}(y_\gamma | M_\gamma p, \sigma^2 I) \\
&= \sum_{\gamma \in \mathcal{Q}} \sum_{\tau \subseteq \gamma} -\log \mathcal{N}(A^+_{\gamma,\tau} y_\tau \mid R_\tau p, \ \sigma^2 \prod_{k \in \gamma \setminus \tau} n_k \cdot D_\gamma D_\gamma^\top) + \mathrm{const.} \\
&= \underbrace{\sum_{\tau \in \mathcal{S}} \sum_{i=1}^{k_\tau} -\log \mathcal{N}(z_{\tau,i} \mid R_\tau p, \ \sigma^2_{\tau,i} D_\gamma D_\gamma^\top) + \mathrm{const.}}_{\mathcal{L}_z(p)}
\end{aligned}$$

where in the final line we rearranged terms using the notation of the theorem statement.

Therefore, $\mathrm{SE}(p)$, $\mathcal{L}_y(p)$ and $\mathcal{L}_z(p)$ all have the same minimizers.

Furthermore, $\mathcal{L}_z(p)$ decomposes over residual measurements as $\mathcal{L}_z(p) = \sum_{\tau \in \mathcal{S}} L_\tau(R_\tau p)$ where $L_\tau(\alpha_\tau) = \sum_{i=1}^{k_\tau} -\log \mathcal{N}(z_{\tau,i} \mid \alpha_\tau, \ \sigma^2_{\tau,i} D_\gamma D_\gamma^\top)$. Therefore, Theorem 1 allows us to minimize each term separately. Algorithm 3 finds $\hat\alpha_\tau$ to minimize $L_\tau(\alpha_\tau)$ for each $\tau \in \mathcal{S}$ using inverse variance weighting. Then, by Theorem 1, the vector $R_\mathcal{S}^+ \hat\alpha$ is a minimizer of $\mathcal{L}_z(p)$, and therefore also a minimizer of $\mathrm{SE}(p)$.

It remains to show that $R_\mathcal{S}^+ \hat\alpha \in \mathrm{row}(M_\mathcal{Q})$. This is true because $R_\mathcal{S}^+ \hat\alpha \in \mathrm{col}(R_\mathcal{S}^+) = \mathrm{row}(R_\mathcal{S}) \subseteq \mathrm{row}(M_\mathcal{Q})$.[3] The final inclusion is true by Lemma 1, since for each $\tau \in \mathcal{S}$ we have $R_\tau = A^+_{\gamma,\tau} M_\gamma$ for some $\gamma \in \mathcal{Q}$. $\qquad\square$

---

[3] In fact $\mathrm{row}(R_\mathcal{S}) = \mathrm{row}(M_\mathcal{Q})$ but we only need the inclusion.

Let us now discuss a generalization of Theorem 3 to the case where noise scales vary across marginal measurements.

**Theorem 5.** *Let $M_{\mathcal{Q}} = (M_\gamma)_{\gamma \in \mathcal{Q}}$ be the query matrix for a multiset $\mathcal{Q}$ of marginals and let $y = (y_\gamma)_{\gamma \in \mathcal{Q}}$ be corresponding noisy marginal measurements with $y_\gamma = M_\gamma p + \mathcal{N}(0, \sigma^2 \mathcal{I})$. Define the scaled query matrix for $\mathcal{Q}$ as $V_{\mathcal{Q}} = (V_\gamma)_{\gamma \in \mathcal{Q}}$ where $V_\gamma = \frac{1}{\sigma_\gamma} M_\gamma$ and the scaled marginal measurements as $v = (v_\gamma)_{\gamma \in \mathcal{Q}}$ where $v_\gamma = \frac{1}{\sigma_\gamma} y_\gamma$. Let $\mathcal{S} = \{\tau \subseteq \gamma : \gamma \in \mathcal{Q}\}$ and for each $\tau \in \mathcal{S}$ let $\gamma_{\tau,i}$ be the ith marginal in $\mathcal{Q}$ containing $\tau$. Let $z_{\tau,i} = A^+_{\gamma_{\tau,i},\tau} y_{\gamma_{\tau,i}}$ be the residual measurement obtained from $\gamma_{\tau,i}$ and let $\Sigma_{\tau,i} = \sigma^2_{\tau,i} D_\tau D_\tau^\top$ be its covariance where $\sigma^2_{\tau,i} = \sigma^2 \prod_{k \in \gamma_{\tau,i} \setminus \tau} n_k$. Then, given any workload of marginal queries $\mathcal{W}$, for each $\gamma \in \mathcal{W}$, the marginal reconstruction $\hat{\mu}_\gamma$ obtained from Algorithm 3 on these residual measurements is equal to $M_\gamma V_{\mathcal{Q}}^+ v$.*

The result follows due to the following Lemma, which shows that $V_{\mathcal{Q}}^+ y$ is an MLE for $p$ given the noisy marginal measurements $y$.

**Lemma 3.** *Let $M_{\mathcal{Q}} = (M_{\gamma_j})_{j=1}^r$ be the query matrix for marginals $\mathcal{Q} = (\gamma_1, \dots, \gamma_r)$, which may include duplicates, and let $y = (y_{\gamma_j})_{j=1}^r$ be corresponding noisy marginal measurements with $y_{\gamma_j} = M_{\gamma_j} p + \mathcal{N}(0, \sigma^2_{\gamma_j} \mathcal{I})$. Define the scaled query matrix as $V_Q = (V_{\gamma_j})_{j=1}^r$ where $V_{\gamma_j} = \frac{1}{\sigma_{\gamma_j}} M_{\gamma_j}$ and the scaled marginal measurements as $v = (v_{\gamma_j})_{j=1}^r$ where $v_{\gamma_j} = \frac{1}{\sigma_{\gamma_j}} y_{\gamma_j}$. Then $V_Q^+ v$ is a MLE of $p$ with respect to noisy measurements $y$.*

*Proof.* The log-likelihood of data vector $p$ under noisy marginal measurement $y_{\gamma_j}$ can be written as

$$\mathcal{L}_{y_{\gamma_j}}(p) = -\frac{1}{2\sigma^2_{\gamma_j}} \left\| y_{\gamma_j} - M_{\gamma_j} p \right\|_2^2 + c_{\gamma_j}$$

where $c_{\gamma_j}$ is a constant. Since the noisy marginal measurements are independent, the log-likelihood of data vector $p$ under noisy marginal measurements $y$ is given by

$$\mathcal{L}_y(p) = -\frac{1}{2} \sum_{j=1}^r \frac{1}{\sigma^2_{\gamma_j}} \left\| y_{\gamma_j} - M_{\gamma_j} p \right\|_2^2 + c$$

where $c$ is a constant. The vector $\hat{p}$ is an MLE of $p$ under noisy marginal measurements $y$ if and only if $\hat{p}$ minimizes the loss function

$$
\begin{aligned}
L_y(p) &= \sum_{j=1}^r \frac{1}{\sigma^2_{\gamma_j}} \left\| y_{\gamma_j} - M_{\gamma_j} p \right\|_2^2 \\
&= \sum_{j=1}^r \left\| \left(\frac{1}{\sigma_{\gamma_j}}\right) y_{\gamma_j} - \left(\frac{1}{\sigma_{\gamma_j}}\right) M_{\gamma_j} p \right\|_2^2 \\
&= \sum_{i=1}^r \left\| v_{\gamma_j} - V_{\gamma_j} p \right\|_2^2 \\
&= \left\| v - V_Q p \right\|_2^2.
\end{aligned}
$$

Since $V_Q^+ v$ minimizes $L_y(p)$, it is an MLE of $p$ under noisy marginal measurements $y$. $\qquad\square$

## E   Computational Complexity

In this section, we analyze the computational complexity of applications of ReM under Gaussian noise. We state and prove the results discussed in Section 4.4. Let us first prove two useful lemmas regarding the time complexity of Alg 5 for multiplying the Kronecker matrix $A = \bigotimes_{i=1}^\ell A_i$ by a vector $x$. Recall that $A_i$ has size $a_i \times b_i$ and $A$ has size $a \times b$ with $a = \prod_{i=1}^\ell a_i$ and $b = \prod_{i=1}^\ell b_i$.

**Lemma 4.** *At iteration $i$, Alg. 5 has the following time complexity:*

*(a) if $A_i$ is an arbitrary matrix, iteration $i$ takes $\mathcal{O}\left( \prod_{j=1}^i a_j \prod_{h=i}^\ell b_h \right)$ time.*

(b) if $A_i = D_{(k)}^+$, then iteration $i$ takes $\mathcal{O}\left(\prod_{j=1}^{i-1} a_j \prod_{h=i}^{\ell} b_h\right)$ time, where $b_i = n_k - 1$.

(c) if $A_i = D_{(k)}$, then iteration $i$ takes $\mathcal{O}\left(\prod_{j=1}^{i} a_j \prod_{h=i+1}^{\ell} b_h\right)$ time, where $a_i = n_k - 1$.

*Proof.* At iteration $i$ of Alg. 5, $A_i$ is multiplied by a matrix $Z$ with size $b_i \times \left(\prod_{j=1}^{i-1} a_j \prod_{h=i+1}^{\ell} b_h\right)$. Then each row in $A_i$ requires $b_i\left(\prod_{j=1}^{i-1} a_j \prod_{h=i+1}^{\ell} b_h\right) = \left(\prod_{j=1}^{i-1} a_j \prod_{h=i}^{\ell} b_h\right)$ scalar multiplications. Since $A_i$ has $a_i$ rows, this yields $\left(\prod_{j=1}^{i} a_j \prod_{h=i}^{\ell} b_h\right)$ multiplications over all rows. This proves $(a)$.

Suppose $A_i = D_{(k)}^+$. Recall that $D_{(k)}^+ = (1/n_k)(1_k u_k^\top - n_k C_k)$. We claim that computing $D_{(k)}^+ v$ for any vector $v$ takes $\mathcal{O}(n_k)$ time. $C_k v$ is a cumulative sum of the elements of $v$ and $u_k^\top v$ is a dot product, both of which take $\mathcal{O}(n_k)$ time to compute. The remaining steps cost $2(n_k - 1)$ multiplications and $n_k - 1$ sums. Thus each column of $Z$ can be multiplied by $A_i$ in $\mathcal{O}(n_k)$ time. Since $Z$ has $\left(\prod_{j=1}^{i-1} a_j \prod_{h=i+1}^{\ell} b_h\right)$ columns, computing $A_i Z$ takes $\mathcal{O}\left(b_i\left(\prod_{j=1}^{i-1} a_j \prod_{k=i+1}^{\ell} b_k\right)\right) = \mathcal{O}\left(\prod_{j=1}^{i-1} a_j \prod_{k=i}^{\ell} b_k\right)$ time, where $b_i = n_k - 1$. This proves $(b)$.

Suppose $A_i = D_{(k)}$. For vector $v$, $D_{(k)} v$ is the difference of consecutive elements of $v$, which takes $\mathcal{O}(n_k)$ time to compute. Thus each column of $Z$ can be multiplied by $A_i$ with $n_k - 1$ operations. Since $Z$ has $\left(\prod_{j=1}^{i-1} a_j \prod_{h=i+1}^{\ell} b_h\right)$ columns, computing $A_i Z$ takes $\mathcal{O}\left(a_i\left(\prod_{j=1}^{i-1} a_j \prod_{k=i+1}^{\ell} b_k\right)\right) = \mathcal{O}\left(\prod_{j=1}^{i} a_j \prod_{k=i+1}^{\ell} b_k\right)$ time, where $a_i = n_k - 1$. This proves $(c)$. $\qquad\square$

**Lemma 5.** *The following hold for Alg. 5:*

(a) *If $a_i \geq b_i$ and either $A_i = D_{(k)}^+$ or $b_i = 1$ for $i = 1, \ldots, \ell$, then Alg. 5 takes $\mathcal{O}(a \cdot \ell)$ time.*

(b) *If $a_i \leq b_i$ and either $A_i = D_{(k)}$ or $a_i = 1$ for $i = 1, \ldots, \ell$, then Alg. 5 takes $\mathcal{O}(b \cdot \ell)$ time.*

*Proof.* Applying Lemma 4, if $A_i = D_{(k)}^+$ then iteration $i$ takes $\mathcal{O}\left(\prod_{j=1}^{i-1} a_j \prod_{h=i}^{\ell} b_h\right)$ time, and, if $b_i = 1$, then iteration $i$ takes $\mathcal{O}\left(\prod_{j=1}^{i} a_j \prod_{h=i+1}^{\ell} b_h\right)$ time. We can bound these terms by $\mathcal{O}\left(\prod_{j=1}^{\ell} a_j\right) = \mathcal{O}(a)$. Summing over all $\ell$ iterations of Alg. 5 yields $\mathcal{O}(\sum_{i=1}^{\ell} a) = \mathcal{O}(a \cdot \ell)$. This proves $(a)$.

If $A_i = D_{(k)}$, then iteration $i$ is $\mathcal{O}\left(\prod_{j=1}^{i} a_j \prod_{h=i+1}^{\ell} b_h\right)$ by Lemma 4 $(c)$. If $a_i = 1$, then iteration $i$ is $\mathcal{O}\left(\prod_{j=1}^{i-1} a_j \prod_{h=i}^{\ell} b_h\right)$ by Lemma 4 $(a)$. We can bound these terms by $\mathcal{O}\left(\prod_{h=1}^{\ell} b_h\right) = \mathcal{O}(b)$. Summing over all $\ell$ iterations of Alg. 5 yields $\mathcal{O}(\sum_{i=1}^{\ell} b) = \mathcal{O}(b \cdot \ell)$. This proves $(b)$. $\qquad\square$

**Theorem 6.** *Let $\mathcal{W}$ be a workload of marginals. Then*

(a) *Reconstructing an answer to marginal $M_\gamma$ for $\gamma \in \mathcal{W}$ takes $\mathcal{O}(|\gamma| n_\gamma 2^{|\gamma|})$ time.*

(b) *The time required for reconstructing an answer to marginal $M_\gamma$ for $\gamma \in \mathcal{W}$ is $o(n_\gamma^{1+\epsilon})$ for any $\epsilon > 0$ as $n_i \to \infty$ for some $i \in \gamma$.*

(c) *GReM-MLE$(\mathcal{W}, \mathcal{S}, z)$ takes $\mathcal{O}(\sum_{\gamma \in W} |\gamma| n_\gamma 2^{|\gamma|})$ time.*

(d) *EMP$(\mathcal{W}, \mathcal{Q}, y)$ takes $\mathcal{O}(\sum_{\gamma \in W} |\gamma| n_\gamma 2^{|\gamma|})$ time.*

(e) *GReM-LNN$(\mathcal{W}, \mathcal{S}, z)$ takes $\mathcal{O}(\sum_{\gamma \in W} |\gamma| n_\gamma 2^{|\gamma|})$ time per round.*

*Proof.* Let us first consider the running time of $A_{\gamma, \tau} z_\tau$ for some $\tau \subseteq \gamma$. Recall that $A_{\gamma, \tau}$ can be written as follows:

$$A_{\gamma, \tau} := \bigotimes_{k \in \gamma} \begin{cases} D_{(k)}^+ & k \in \tau \\ (1/n_k) 1_k & k \in \gamma \setminus \tau \end{cases}$$

Since $A_{\gamma, \tau}$ satisfies the conditions of Lemma 5 and has $n_\gamma$ rows, computing $A_{\gamma, \tau} z_\tau$ takes $\mathcal{O}(|\gamma| n_\gamma)$ time. Recall from Proposition 1 that reconstructing an answer to marginal $M_\gamma$ is given by

$\sum_{\tau \in \mathcal{S}, \tau \subseteq \gamma} A_{\gamma, \tau} y_\tau$. The number of terms in the summation is at most $2^{|\gamma|}$, so the total running time of reconstructing an answer to $M_\gamma$ is $\mathcal{O}(|\gamma| n_\gamma 2^{|\gamma|})$. This proves $(a)$.

For $(b)$, let $\epsilon > 0$ and consider the following quotient:

$$\frac{|\gamma| n_\gamma 2^{|\gamma|}}{|\gamma| n_\gamma^{1+\epsilon}} = \frac{2^{|\gamma|}}{n_\gamma^\epsilon} = \frac{2^{|\gamma|}}{\prod_{i \in \gamma} n_i^\epsilon}.$$

Taking the limit as $n_i \to \infty$, the quotient tends to zero and we obtain the desired result.

With GReM-MLE, each residual query $R_\tau, \tau \in \mathcal{S}$ can have multiple measurements $y_{\tau,1}, \ldots, y_{\tau, k_\tau}$ but with proportional covariances. For each $\tau \in \mathcal{S}$, we combine the measurements using inverse variance weighting to obtain $\hat{\alpha}_\tau$. We then reconstruct the marginals $M_\gamma$ for $\gamma \in \mathcal{W}$ using the residual answers $\hat{\alpha}_\tau$ for $\tau \in \mathcal{S}$. By $(a)$, the running time is $\mathcal{O}(\sum_{\gamma \in W} |\gamma| n_\gamma 2^{|\gamma|})$. This proves $(c)$.

The efficient marginal pseudoinversion, given in Alg. 4, first decomposes marginals and then applies GReM-MLE. Let $\mathcal{Q}$ be the multiset of measured marginals and $\mathcal{W}$ be the workload of marginals to answer. Let $\mathcal{W}^\downarrow$ denote the downward closure of $\mathcal{W}$. We assume that $\mathcal{Q}$ is a consists of elements of $\mathcal{W}^\downarrow$ and each $\gamma \in \mathcal{W}^\downarrow$ appears in $\mathcal{Q}$ at most $b$ times. For each $\gamma \in \mathcal{Q}$, we decompose the marginal measurements into residual measurements by computing $A_{\gamma, \tau}^+ y_\gamma$ for each $\tau \subseteq \gamma$. By Lemma 5 $(b)$, computing $A_{\gamma, \tau}^+ y_\gamma$ takes $\mathcal{O}(|\gamma| n_\gamma)$ time. Then the running time of decomposing the marginal measurements is $\mathcal{O}(\sum_{\gamma \in \mathcal{Q}} |\gamma| n_\gamma 2^{|\gamma|})$. From $(c)$, the running time of GReM-MLE is $\mathcal{O}(\sum_{\gamma \in \mathcal{W}} |\gamma| n_\gamma 2^{|\gamma|})$. Given that the running time of decomposition is at most a multiple of the running time of GReM-MLE, $\mathcal{O}(\sum_{\gamma \in \mathcal{W}} |\gamma| n_\gamma 2^{|\gamma|})$. This proves $(d)$.

Let us turn to the running time of GReM-LNN. Let $\mathcal{W}^\downarrow$ be the downward closure of workload $\mathcal{W}$. The dual ascent algorithm for GReM-LNN (Alg. 6) consists of three steps each round requiring matrix multiplications: computing $\hat{\alpha}_\tau$ for $\tau \in \mathcal{S}$, computing $\hat{\alpha}_{\tau'}$ for unmeasured $\tau' \in \mathcal{W}^\downarrow \setminus \mathcal{S}$, and reconstructing answers to marginals $M_\gamma$ for $\gamma \in \mathcal{W}$.

First consider the case where $\tau \in S$. Recall that in this case $\hat{\alpha}_\tau = \left(\sum_{i=1}^{k_\tau} K_{\tau,i}^{-1}\right)^{-1} \left(\sum_{i=1}^{k_\tau} K_{\tau,i}^{-1} y_{\tau,i} + \sum_{\gamma \supseteq \tau} A_{\gamma,\tau}^T \lambda_\gamma\right)$, where $K_{\tau,i} = \sigma_\tau^2 D_\tau D_\tau^T$. We can rewrite $\hat{\alpha}_\tau$ as follows:

$$\hat{\alpha}_\tau = \left(\sum_{\gamma \supseteq \tau} \sigma_\tau^{-2}\right)^{-1} \sum_{\gamma \supseteq \tau} \sigma_\tau^{-2} y_{\tau,i} + \left(\sum_{\gamma \supseteq \tau} \sigma_\tau^{-2}\right) D_\tau D_\tau^T \sum_{\gamma \supseteq \tau} A_{\gamma,\tau}^T \lambda_\gamma.$$

The left summand requires no matrix multiplications and does not depend on $\lambda$. Then computing $A_{\gamma,\tau}^T \lambda_\gamma$ takes $\mathcal{O}(|\gamma| n_\gamma)$ time. The right summand is obtained by computing $A_{\gamma,\tau}^T \lambda_\gamma$ for each $\gamma \supseteq \tau$. Then computing $\hat{\alpha}_\tau$ for $\tau \in \mathcal{S}$ takes $\mathcal{O}(\sum_{\gamma \supseteq \tau} |\gamma| n_\gamma)$ time.

Now consider the case where $\tau \in \mathcal{W}^\downarrow \setminus S$. Then $\hat{\alpha} = -(1/2)(A_{\tau,\tau}^T A_{\tau,\tau})^{-1} \sum_{\gamma \supseteq \tau} A_{\gamma,\tau}^T \lambda_\gamma$. As with the prior case, the desired term requires computing $A_{\gamma,\tau}^T \lambda_\gamma$ for each $\gamma \supseteq \tau$. Then computing $\hat{\alpha}_\tau$ for $\tau \in \mathcal{W}^\downarrow \setminus \mathcal{S}$ is $\mathcal{O}(\sum_{\gamma \supseteq \tau} |\gamma| n_\gamma)$.

Combing these results, computing $\hat{\alpha}_\tau$ for $\tau \in \mathcal{W}^\downarrow$ is $\mathcal{O}(\sum_{\tau \in \mathcal{W}^\downarrow} \sum_{\gamma \supseteq \tau} |\gamma| n_\gamma)$. Observe that for each $\gamma \in \mathcal{W}$, there are $2^{|\gamma|}$ terms in the summation. By indexing the summation in terms of $\gamma$, we obtain that computing $\hat{\alpha}$ is $\mathcal{O}(\sum_{\gamma \in W} |\gamma| n_\gamma 2^{|\gamma|})$. The remaining step of GReM-LNN is to reconstruct answers to marginals $M_\gamma$ for $\gamma \in \mathcal{W}$. By $(a)$, the running time is $\mathcal{O}(\sum_{\gamma \in W} |\gamma| n_\gamma 2^{|\gamma|})$. This proves $(e)$.

$\square$

## F   GReM-LNN Implementation

Recall that GReM-LNN solves the following convex program:

$$\min_\alpha \sum_{\tau \in \mathcal{S}} \sum_i (\alpha_\tau - z_{\tau,i})^\top K_{\tau,i}^{-1} (\alpha_\tau - z_{\tau,i}) \quad \text{s.t.} \sum_{\tau \subseteq \gamma} A_{\gamma,\tau} \alpha_\tau \geq 0, \quad \forall \gamma \in \mathcal{W} \tag{8}$$

---

**Algorithm 6** GReM-LNN Dual Ascent

---

**Input:** Marginal workload $\mathcal{W}$, residual workload $\mathcal{S}$, residual measurements $z$, rounds $T$, step size $s$, Lagrangian initialization $\lambda$, regularization weight $\eta$

1: Initialize $\lambda_\gamma = \lambda$ for $\gamma \in \mathcal{W}$
2: **for** $t = 1, \ldots, T$ **do**
3:      Set $\alpha_\tau = \left( \sum_{i=1}^{k_\tau} K_{\tau,i} \right)^{-1} \left( \sum_{i=1}^{k_\tau} K_{\tau,i}^{-1} y_{\tau,i} - \sum_{\gamma \supseteq \tau} A_{\gamma\tau}^\top \lambda_\gamma \right)$ for $\tau \in \mathcal{S}$
4:      Set $\alpha_\tau = -1/2\eta \left( A_{\tau\tau}^\top A_{\tau\tau} \right)^{-1} \left( \sum_{\gamma \supseteq \tau} A_{\gamma\tau}^\top \lambda_\gamma \right)^\top$ for $\tau \in \mathcal{W}^\downarrow \setminus \mathcal{S}$
5:      Calculate $\mu_\gamma(\alpha) = \sum_{\tau \subseteq \gamma} A_{\gamma\tau} \alpha_\tau$ for $\gamma \in \mathcal{W}$
6:      Update $\lambda_\gamma = \min\{\lambda_\gamma + s\mu_\gamma(\alpha), 0\}$ for $\gamma \in \mathcal{W}$

---

for $K_{\tau,i} = 2^{|\tau|} D_\tau D_\tau^T$. Observe that the program in Eq. 8 only depends on unmeasured residuals in $\mathcal{W}^\downarrow$ through the local non-negativity constraint. To make this problem more tractable and the solution more stable, we introduce a regularization term to limit the contribution of unmeasured residuals to reconstructed marginals:

$$
\min_\alpha \sum_{\tau \in \mathcal{S}} \sum_i (\alpha_\tau - z_{\tau,i})^\top K_{\tau,i}^{-1} (\alpha_\tau - z_{\tau,i}) + \eta \sum_{\nu \in \mathcal{W}^\downarrow \setminus \mathcal{S}} \|A_{\nu\nu} \alpha_\nu\|_2^2
$$
$$
\text{s.t.} \sum_{\tau \subseteq \gamma} A_{\gamma,\tau} \alpha_\tau \geq 0, \quad \forall \gamma \in \mathcal{W}
\tag{9}
$$

Note that the introduction the regularization term in Eq. (9) is only relevant to the underdetermined case, since, otherwise, $\mathcal{W}^\downarrow \subseteq \mathcal{S}$. To solve the program in Eq. (9), we use an iterative dual ascent algorithm described in pseudocode in Alg. 6.

Let us now show that Alg. 6 is correctly specified. Let us denote the objective as $f(\alpha) = \sum_{\tau \in \mathcal{S}} \sum_{i=1}^{k_\tau} (\alpha_\tau - z_{\tau,i})^\top K_{\tau,i}^{-1} (\alpha_\tau - z_{\tau,i}) + \eta \sum_{\nu \in \mathcal{W}^\downarrow \setminus \mathcal{S}} \|A_{\nu\nu} \alpha_\nu\|_2^2$ and the constraint as $\mu(\alpha) = (\mu_\gamma(\alpha))_{\gamma \in \mathcal{W}} = (\sum_{\tau \subseteq \gamma} A_{\gamma\tau} \alpha_\tau)_{\gamma \in \mathcal{W}} \geq 0$. Then the Lagrangian function is given by

$$
\mathcal{L}(\alpha, \lambda) = f(\alpha) + \lambda^\top \mu(\alpha)
$$
$$
= \sum_{\tau \in \mathcal{S}} \sum_{i=1}^{k_\tau} (\alpha_\tau - z_{\tau,i})^\top K_{\tau,i}^{-1} (\alpha_\tau - z_{\tau,i}) + \eta \sum_{\nu \in \mathcal{W}^\downarrow \setminus \mathcal{S}} \|A_{\nu\nu} \alpha_\nu\|_2^2 + \sum_{\gamma \in \mathcal{W}} \lambda_\gamma^\top \sum_{\tau \subseteq \gamma} A_{\gamma\tau} \alpha_\tau
$$

where $\lambda = (\lambda_\gamma)_{\gamma \in \mathcal{W}}$ is the dual variable or Lagrangian multiplier and is constrained such that $\lambda \leq 0$. The dual function is given by $g(\lambda) = \min_\alpha \mathcal{L}(\alpha, \lambda)$ and the dual problem is given by $\max_{\lambda \leq 0} g(\lambda)$. Under suitable regularity conditions, the optimal value of the dual problem is equivalent to the optimal value of the primal problem. We can solve both by maximizing the dual function $g$ to obtain $\lambda^*$ and then minimizing the Lagrangian $\mathcal{L}(\alpha, \lambda^*)$ with respect to $\alpha$ to obtain $\alpha^*$.

We can solve for each $\alpha_\tau^*$ in closed form for $\tau \in \mathcal{W}^\downarrow$. Minimizing the Lagrangian $\mathcal{L}(\alpha, \lambda^*)$ with respect to $\alpha$ corresponds to minimizing an unconstrained quadratic objective and can be solved separately for each $\tau$. To see this, let us fix $\lambda$ and solve for the critical point of $\mathcal{L}(\alpha, \lambda)$. If $\tau \in \mathcal{S}$, then gradient of $\mathcal{L}$ with respect to $\alpha_\tau$ is given by

$$
\nabla_{\alpha_\tau} \mathcal{L}(\alpha, \lambda) = \sum_{i=1}^{k_\tau} K_{\tau,i}^{-1} (\alpha_\tau - z_{\tau,i}) + \sum_{\gamma \supseteq \tau} A_{\gamma\tau}^\top \lambda_\gamma.
$$

Setting this to zero and solving for $\alpha_\tau^*$ yields

$$
\alpha_\tau^* = \left( \sum_{i=1}^{k_\tau} K_{\tau,i}^{-1} \right)^{-1} \left( \sum_{i=1}^{k_\tau} K_{\tau,i}^{-1} z_{\tau,i} - \sum_{\gamma \supseteq \tau} A_{\gamma\tau}^\top \lambda_\gamma \right).
$$

---

**Algorithm 7** Scalable MWEM

---

**Input:** Marginal workload $\mathcal{W}$, privacy budget $(\epsilon, \delta)$, initialization parameter $\alpha$

1: Choose $\rho$ such that $\min_{\alpha>1} \frac{\exp((\alpha-1)(\alpha\rho-\epsilon))}{\alpha-1} \left(1 - \frac{1}{\alpha}\right)^{\alpha} = \delta$

2: Set $\sigma_0^2, \sigma^2 = \frac{1}{2\alpha\rho}, \frac{T}{(1-\alpha)\rho}$

3: Initialize measurements $y = \{M_\emptyset p + \xi_0\}$ with $\xi_0 \sim \mathcal{N}(0, \sigma_0^2 I)$ and multiset $\mathcal{Q} = \{\emptyset\}$

4: Initialize $(\hat{\mu}_\gamma)_{\gamma \in \mathcal{W}} = \text{EMP}(\mathcal{W}, \mathcal{Q}, y)$

5: **for** $t = 1, \ldots, T$ **do**

6:     Select $\gamma_t$ with the exponential mechanism using $\frac{(1-\alpha)\rho}{2T}$ budget according to

$$\text{Score}(p, \gamma, Y) = \|M_\gamma p - \hat{\mu}_\gamma\|_1 \ \forall \gamma \in \mathcal{W}$$

7:     Measure $y_t = M_{\gamma_t} p + \xi_t$ where $\xi_t \sim \mathcal{N}(0, \sigma^2 I)$ and set $\mathcal{Q} = \mathcal{Q} \cup \{\gamma_t\}$

8:     Reconstruct $(\hat{\mu}_\gamma)_{\gamma \in \mathcal{W}} = \text{EMP}(\mathcal{W}, \mathcal{Q}, y)$

    **return** noisy answers $(\hat{\mu}_\gamma)_{\gamma \in \mathcal{W}}$, noisy measurements $y$

---

Now, suppose $\tau \in \mathcal{W}^{\downarrow} \setminus \mathcal{S}$. Then gradient of $\mathcal{L}$ with respect to $\alpha_\tau$ is given by

$$\nabla_{\alpha_\tau} \mathcal{L}(\alpha, \lambda) = 2\eta \alpha_\tau^\top A_{\tau\tau}^\top A_{\tau\tau} + \sum_{\gamma \supseteq \tau} A_{\gamma\tau}^\top \lambda_\gamma.$$

Setting this to zero and solving for $\alpha_\tau^*$ yields

$$\alpha_\tau^* = -\nicefrac{1}{2\eta} \left(A_{\tau\tau}^\top A_{\tau\tau}\right)^{-1} \left(\sum_{\gamma \supseteq \tau} A_{\gamma\tau}^\top \lambda_\gamma\right)^\top.$$

To update $\lambda$, we set $\lambda^* = \min\{\lambda + t\mu(\alpha^*), 0\}$ where $t > 0$ is the step size. This can be seen as projected gradient ascent on $g(\lambda)$ since $\mu(\alpha^*) = \nabla_\lambda \mathcal{L}(\alpha^*, \lambda) = \nabla_\lambda g(\lambda)$.

## G   Scalable MWEM with pseudoinverse reconstruction

The multiplicative weights exponential mechanism (MWEM) [15] is a canonical data-dependent mechanism that maintains a model $\hat{p}$ of the data distribution $p$ that is improved iteratively by adaptively measuring marginal queries that are poorly approximated by the current model $\hat{p}$. MWEM has served as the foundation for many related data-dependent mechanisms. A limitation of MWEM-style algorithms is that representing $\hat{p}$, even implicitly, does not scale to high-dimensional data domains without adopting parametric assumptions. In this section, we propose an MWEM-style algorithm called Scalable MWEM (Alg. 7) that employs a standard reconstruction approach, the pseudoinverse of the measured marginal queries, but scales to high-dimensional data domains.

In general, the pseudoinverse is infeasible as a reconstruction method for large data domains. Computing the pseudoinverse $Q^+$ of an arbitrary query matrix $Q$ scales exponentially in the number of attributes and linearly in size of the data vector. Moreover, even storing the reconstructed data vector $\hat{p} = Q^+ y$ from noisy answers $y$ in memory presents a limitation in practice. Scalable MWEM overcomes this computational hurdle by measuring marginals with isotropic noise and utilizing the efficient marginal pseudoinverse (Alg. 4).

Scalable MWEM initializes by using a predetermined fraction of the privacy budget to measure the total query i.e. the 0-way marginal that counts the number of records in the dataset. Let $\mathcal{W}$ be a workload of marginals e.g. all 3-way marginals. Then, for a fixed number of rounds, Scalable MWEM privately selects a marginal $\gamma \in \mathcal{W}$ that is poorly approximated by the pseudoinverse of the current measurements using the exponential mechanism. The selected marginal is measured with isotropic Gaussian noise and utilizes the efficient marginal pseudoinverse to reconstruct answers to marginals in $\mathcal{W}$. Being a full query answering mechanism rather than just a reconstruction method, let us show that Scalable MWEM satisfies differential privacy.

**Theorem 7.** *Scalable MWEM satisfies $(\epsilon, \delta)$-DP.*

*Proof.* We will refer to Algorithm 7 as $\mathcal{M}$. Note that $\mathcal{M}$ selects a parameter $\rho$ such that $\delta = \min_{\alpha>1} \frac{\exp((\alpha-1)(\alpha\rho-\epsilon))}{\alpha-1}\left(1 - \frac{1}{\alpha}\right)^\alpha$. By proposition 4, it suffices to show that $\mathcal{M}$ satisfies $\rho$-zCDP, then it also satisfies $(\epsilon, \delta)$-DP. In the initialization step, $\mathcal{M}$ measures $M_\emptyset p$ with the Gaussian mechanism using the noise scale $\sigma_o^2 = \frac{1}{2\alpha\rho}$. The query $M_\emptyset p$ is the total query, so it has an $\ell_2$ sensitivity of 1 and therefore by proposition 5, this measurement satisfies $\frac{1}{2\sigma_o^2} = \frac{2\alpha\rho}{2} = \alpha\rho$-zCDP. In each round, $\mathcal{M}$ runs the exponential mechanism such that it satisfies $\frac{(1-\alpha)\rho}{2T}$-zCDP. Also in each round, $\mathcal{M}$ runs the Gaussian mechanism to measure a marginal query with noise scale $\sigma^2 = \frac{T}{(1-\alpha)\rho}$. All marginal queries have an $\ell_2$ sensitivity of 1 so again by proposition 5, this measurement satisfies $\frac{1}{2\sigma_o^2} = \frac{(1-\alpha)\rho}{2T}$-zCDP. By the adaptive composition result given in proposition 8, the overall mechanism satisfies $\alpha\rho + T(\frac{(1-\alpha)\rho}{2T} + \frac{(1-\alpha)\rho}{2T}) = \rho$-zCDP and also $(\epsilon, \delta)$-DP. $\qquad\square$

## H  Experiment Details

**Datasets.** In general, we follow the preprocessing steps described in [7]. All attributes in the datasets are discrete. We identify the data domain by inferring the possible values for each attribute from the observed values for each attribute.

Titanic [23] contains 9 attributes, $1,304$ records, and has data vector size $8.9 \times 10^7$. Adult [24] contains 14 attributes, $48,842$ records, and has data vector size $9.8 \times 10^{17}$. Salary [25] contains 9 attributes, $135,727$ records, and has data vector size $1.3 \times 10^{13}$. Nist-Taxi [26] has 10 attributes, $223,551$ records, and has data vector size $1.9 \times 10^{13}$.

**Compute Environment.** All experiments were run on an internal compute cluster with two CPU cores and 20GB of memory.

**GReM-LNN Hyperparameters.** For the ResidualPlanner experiments in Section 5.1, we set the hyperparameters as follows: the maximum number of rounds $T = 4000$, the Lagrangian initialization parameter $\lambda = -1$, and the step size $s = 0.1$. For the Scalable MWEM experiments in Section 5.2, we set the hyperparameters as follows: the maximum number of rounds $T = 1000$, the Lagrangian initialization parameter $\lambda = -1$, the step size $s = 0.02$, and regularization weight $\eta = 40$. For all experiments, if Alg. 6 fails, we divide the step size by $\sqrt{10}$ and rerun until convergence. We additionally impose a time limit of 24H on a given run of Alg. 6.

## I  Additional Experiments

In this section, we detail additional experimental results. For the ResidualPlanner experiment, we report $\ell_2$ workload error for the reconstruction methods. For the Scalable MWEM experiment, we report $\ell_2$ workload error for the reconstruction methods as well as whether or not Private-PGM successfully ran across various settings.

## I.1 Additional ResidualPlanner Experiments

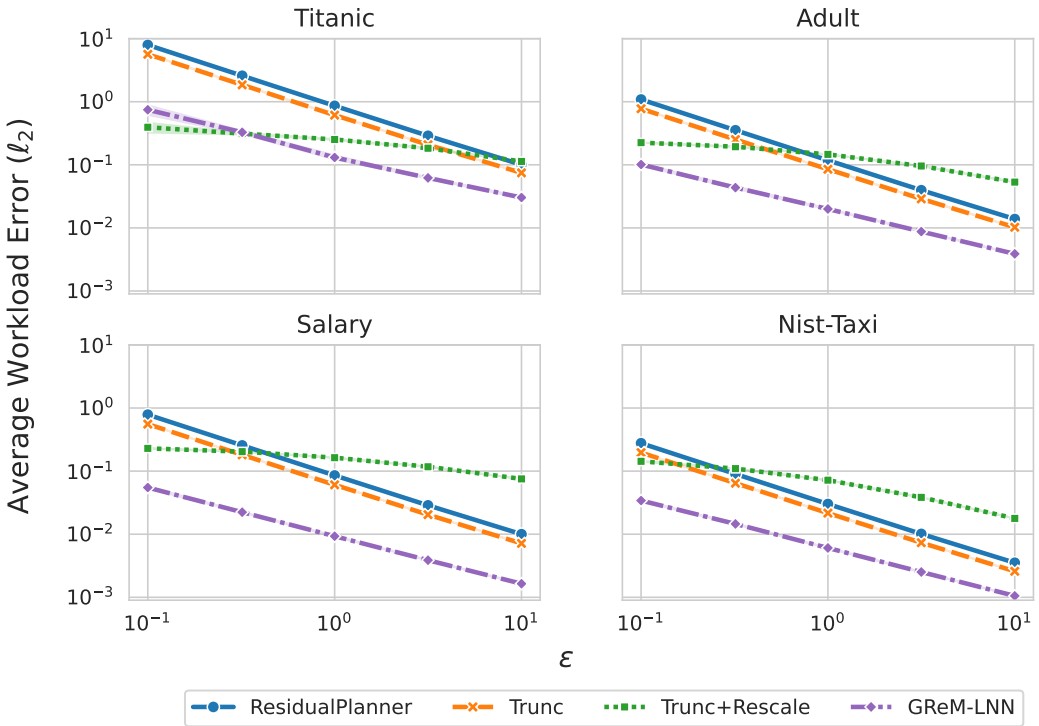

Figure 4: Average $\ell_2$ workload error on all 3-way marginals across five trials and privacy budgets $\epsilon \in \{0.1, 0.31, 1, 3.16, 10\}$ and $\delta = 1 \times 10^{-9}$ for ResidualPlanner.

## I.2  Additional MWEM Experiments

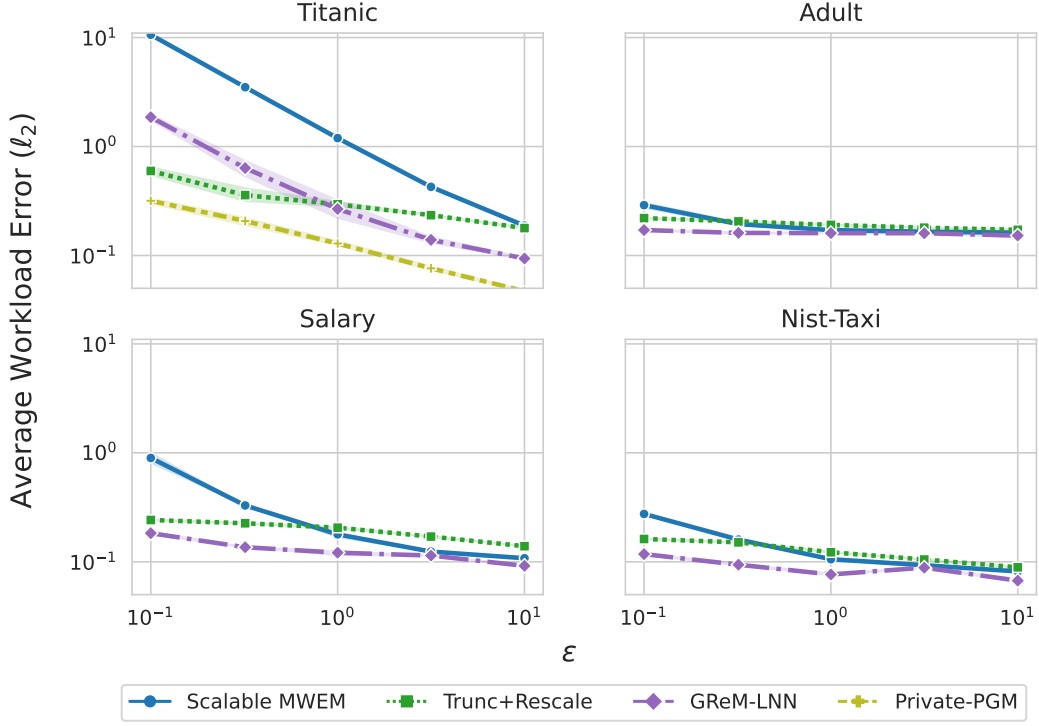

Figure 5: Average $\ell_2$ workload error on all 3-way marginals across five trials and privacy budgets $\epsilon \in \{0.1, 0.31, 1, 3.16, 10\}$ and $\delta = 1 \times 10^{-9}$ for Scalable MWEM with 30 rounds of measurements.

| Dataset | Rounds | Trials Total | Trials Completed | Trials >24H | Trials Out-of-Memory |
|---------|--------|--------------|------------------|-------------|----------------------|
| Titanic | 10 | 25 | 25 | 0 | 0 |
|         | 20 | 25 | 25 | 0 | 0 |
|         | 30 | 25 | 25 | 0 | 0 |
| Adult   | 10 | 25 | 0 | 25 | 0 |
|         | 20 | 25 | 14 | 8 | 3 |
|         | 30 | 25 | 0 | 0 | 25 |
| Salary  | 10 | 25 | 11 | 14 | 0 |
|         | 20 | 25 | 0 | 0 | 25 |
|         | 30 | 25 | 0 | 0 | 25 |
| Nist-Taxi | 10 | 25 | 0 | 0 | 25 |
|         | 20 | 25 | 0 | 0 | 25 |
|         | 30 | 25 | 0 | 0 | 25 |

Table 2: Completion results of running Private-PGM by setting for the Scalable MWEM experiment. Failure is broken down by exceeding the 24H time limit or exceeding the available memory (20GB).

