# OpenReview forum: "Efficient and Private Marginal Reconstruction with Local Non-Negativity"
_NeurIPS.cc/2024/Conference — NeurIPS 2024 poster_

### Official Review · Reviewer_JUY8 · 2024-07-12

**Soundness:** 3
**Presentation:** 3
**Contribution:** 2
**Rating:** 3
**Confidence:** 4

**Summary:**

This paper introduces residuals-to-marginals (ReM), a novel approach to constructing marginals from input data residuals while maintaining differential privacy. ReM operates by first estimating true residuals from noisy ones, then transforming these estimates into target marginals. The authors propose two variants of ReM: ReM-LNN, which enhances accuracy by enforcing non-negativity constraints on the resulting marginals, and GReM-MLE, a more efficient version for scenarios involving Gaussian noise injection to residuals, leveraging a closed-form solution. The paper also presents an extension of ReM that incorporates the MWEM. This extension identifies residuals that are useful for marginal construction, offering an end-to-end solution when residual choices are not predetermined. Experimental results demonstrate that ReM significantly reduces errors compared to baseline methods in constructing differentially private marginals. ReM also demonstrates good scalability, whereas PrivatePGM, a baseline method, failed to run for large data domains.

**Strengths:**

1. This paper presents ReM, a new approach to constructing marginals from residuals.

2. ReM reduces errors compared to some baseline methods.

3. ReM can be applied as a post-processing method to improve the accuracy of some other methods.

**Weaknesses:**

1. The paper compares ReM against MWEM and Private-PGM [12], but omits the state-of-the-art methods such as AIM [7] and PrivMRF [19]. Both AIM and PrivMRF have been demonstrated to significantly outperform MWEM and Private-PGM. As a consequence, it is unclear whether ReM really advances the state of the art.

2. It would be helpful to explicitly define k_tau in Line 1 of Algorithm 2. I do not see where this paper defines k_tau. It seems that for each attribute set tau, the algorithm injects noise k_tau times, resulting in k_tau marginals, ranging from y_{tau, 1} to y_{tau, k_r}. Why is it necessary to inject noise multiple times for each marginal?

**Questions:**

1. Please include the derivation of L_tau in Line 216, Page 5. It appears to represent the probability of generating y_{tau, i} given alpha, over the randomness of Gaussian noise. Additionally, please provide a detailed argument for the closed-form solution mentioned in Line 218.

2. Could you clarify why "when the marginal is observed with isotropic noise, the corresponding noisy residuals are independent"? (Line 239, Page 6)

**Limitations:**

Yes.

---

> ### Author Rebuttal · Authors · 2024-08-07
>
> We thank the reviewer for providing feedback on our paper and suggesting improvements to clarify the presentation. Below, we address various concerns raised in the review.
>
> Since ReM is a reconstruction method, the appropriate comparison is not with full end-to-end synthetic data or query answering mechanisms such as AIM and PrivMRF but rather with the reconstruction methods these mechanisms use.
> Both AIM and PrivMRF use Private-PGM as part of their reconstruction steps, and we include Private-PGM in our experiments.
> Regarding Scalable MWEM, we use this end-to-end mechanism to choose which marginals to measure but compare Scalable MWEM's reconstruction method (discussed in Appendix D) to ReM with Local Non-negativity (GReM-LNN) and Private-PGM in our experiments.
>
> ReM takes an arbitrary number $k_\tau$ of measurements of each residual $R_\tau \in \mathcal{S}$ as input. We will be more explicit about $k_\tau$ when it is introduced in the text on Line 189.
>
> Regarding why a residual query might be measured multiple times, please see the global response.
>
> The loss function $L_\tau$ from Line 216 comes from using the negative log-likelihood as the loss function (as described in Line 190) for the noise model described in Lines 212 and 213, i.e., with $k_\tau$ independent Gaussian random variables $y_{\tau, i}$ with covariance $\Sigma_{\tau, i}$, so that
> $$
> L_\tau(\alpha_\tau) = -\sum_{i=1}^{k_\tau} \log p(y_{\tau, i} | R_\tau p = \alpha_{\tau})
> $$
> with $p(y_{\tau,i} | R_\tau p = \alpha_\tau) = \mathcal{N}(y_{\tau,i}; \alpha_\tau, \Sigma_{\tau,i})$.
> We solve for $\hat \alpha_\tau$ in closed-form by finding the critical point of $L_\tau$. In the revision, we will include the derivation of the closed-form solution to $\hat \alpha_\tau$ in an appendix.
>
> Lines 238-240 are an informal statement of Theorem 2. To clarify this, we will specify that $z_\tau$ is a ''noisy residual'' in the statement of Theorem 2.

---

### Official Review · Reviewer_r5Ym · 2024-07-13

**Soundness:** 3
**Presentation:** 3
**Contribution:** 3
**Rating:** 6
**Confidence:** 3

**Summary:**

The paper studies the problem of reconstructing a data distribution from noisy measurements of a set of carefully selected marginal queries and using the reconstructed data distribution to answer a set of workload queries. The difficulty in the problem lies in the high dimensionality of the data distribution, which in turn requires measuring and inverting an exponential number of queries in the worst case. In this paper, the authors propose to
1. translate the marginal workload queries to their equivalent residual queries representation, and compute their noisy measurements
2. exploiting the structure of residual query matrix to perform efficient pseudoinverse via Kronecker product
3. perform reconstruction via an optimization problem that has solution $=$ the product of pseudoinverse matrix and noisy measurements, where additional non-negativity constraints on the solution are enforced as post-processing

 Compared to prior approaches for marginal reconstruction, the proposed approaches enjoy improved efficiency as the Kronecker product matrix is efficiently invertible. The technique of converting the query matrix to its residual representation is further extended to the MWEM algorithm to allow efficient computation and representation of reconstructed marginals under the high-dimensional data. Experiments confirm that the proposed reconstruction and scalable MWEM algorithms allow improved query-answering performance when used on top of existing mechanisms that are solely based on noisy measurements rather than reconstructed data distribution.

**Strengths:**

- An interesting idea of using the residual basis of query matrix to efficiently represent and compute pseudoinverse matrix for marginal reconstruction.
- Based on the observation, two interesting new algorithms are proposed: efficient marginal reconstruction both via reconstructing data distribution, and scalable MWEM on high-dimensional data via residual computation and representation of marginal queries.
- Strong experimental performance when used as post-processing mechanisms on top of existing query-answering mechanisms that are solely based on noisy measurements rather than reconstructed data distribution.

**Weaknesses:**

1. One concern is regarding the discussion of computation complexity of the proposed method. The main advantage of the proposed algorithms is that they "will not have exponential complexity" (line 46). However, as the proposed algorithms still involve some enumerating operations on all possible attribute subsets (e.g., lines 9-11 in algorithm 3), it is not immediately clear why they "will not have exponential complexity". It would also be helpful if the authors could clarify what they mean by exponential complexity.

2. Relevant to the previous question, there are well-known hardness results of generating synthetic data under differential privacy [a], that talk about the necessity of exponential runtime for accurate reconstruction of data distribution. Such lower bounds should be discussed in detail to understand the results of this paper, in terms of the saved computation complexity.

3. The comparison is mainly done with regard to Residual planner [6] and PGM [12], while several other reconstruction methods such as PrivBayes [8], GEM [9], RAP [10], and RAP++ [11] are not considered in the comparison. Such comparisons are useful for understanding the performance of the proposed reconstruction algorithm, and for understanding the contributions of the paper. Alternatively, it would be useful if the authors could further clarify why such algorithms are not considered in the comparison.


[a] Ullman, Jonathan, and Salil Vadhan. "PCPs and the hardness of generating private synthetic data." Theory of Cryptography Conference. Berlin, Heidelberg: Springer Berlin Heidelberg, 2011.

**Questions:**

1. Why would not the proposed algorithms have exponential complexity (weakness 1)? And how does this relate to the known hardness results (weakness 2)?

2. Could the authors explain more about the lack of comparison with other reconstruction methods (weakness 3)?

---

> ### Author Rebuttal · Authors · 2024-08-07
>
> We thank the reviewer for providing feedback on our paper and suggesting improvements to clarify the presentation. Below, we address various concerns raised in the review.
>
> Regarding time complexity, please see the global response for specific running time bounds, which we will include in the revised paper. Regarding ``exponential complexity'', we will also clarify this. Our goal is to bound the running time of the reconstruction problem in terms of the size of its own inputs and outputs. The inputs are the noisy residual measurements, and the outputs are the reconstructed marginals. The results we state in the global response show that our algorithms take time that is polynomial (and subquadratic) in these size parameters, and thus avoid exponential complexity. As a concrete example, suppose the workload is all 2-way marginals over $d$ attributes, all attributes have domain size $m$, and the measurements are taken from ResidualPlanner. Then the input size and output size are both ${d \choose 2} m^2$. Our methods are polynomial (subquadratic) in ${d \choose 2} m^2$ (and thus in $d$ and $m$), while a method that reconstructs the full data vector will take $\Omega(m^d)$ time.
>
> Thank you for the comment about hardness results. We will add a discussion of complexity results to our paper. Briefly, the theorem of Ullman and Vadhan [a] does not apply because our mechanism does not output a synthetic data set. Indeed, as noted by Vadhan [b], ''the requirement that the mechanism produces a synthetic dataset cannot be removed'' from this hardness result. This is because the class of queries used in the result (two-way marginals) can be answered with vanishing error in polynomial time by a mechanism (e.g., the Laplace mechanism) that directly answers the queries without producing synthetic data. We will also add a discussion of other computational hardness results for query answering. In general, these results rely on worst-case query classes constructed from cryptographic primitives and not natural query classes such as marginals, so we don't expect them to apply to our practical use cases. Vadhan [b] highlights it as an open problem to give query-answering hardness results for natural query classes.
>
> Regarding comparisons, our goal in this paper is to study the reconstruction subproblem and develop general-purpose solutions. Among the alternative algorithms, only Private-PGM is a self-contained reconstruction method. The other algorithms (PrivBayes, RAP, GEM, etc.) are full query-answering or synthetic data mechanisms, which include some approach to reconstruction in the context of their mechanism. In several cases, such as RAP, a generic reconstruction approach can be extracted. RAP uses a neural network to represent the solution space and gradient descent variants to train it. However, to the best of our knowledge, due to the parametric modeling assumptions of these mechanisms, reconstruction requires solving a non-convex optimization problem. Thus the final result will be dependent both on how well the data distribution meets the parametric modeling assumptions, and on how well one is able to solve the non-convex optimization problem in practice. For the purpose of studying reconstruction in a self-contained manner, we choose in this paper to restrict to Private-PGM, which is the only other method that is general purpose and has predictable output due to solving a convex optimization problem to optimality. In-depth empirical comparisons of different reconstruction methods in the context of full mechanisms, including the impacts of non-convexity, etc., is an interesting avenue for future work.
>
> [a] Ullman, Jonathan, and Salil Vadhan. "PCPs and the hardness of generating private synthetic data." Theory of Cryptography Conference. Berlin, Heidelberg: Springer Berlin Heidelberg, 2011.
>
> [b] Vadhan, Salil. The Complexity of Differential Privacy. In: Lindell, Y. (eds) Tutorials on the Foundations of Cryptography. Information Security and Cryptography. Springer, 2017.

---

### Official Review · Reviewer_a9Gz · 2024-07-13

**Soundness:** 3
**Presentation:** 2
**Contribution:** 3
**Rating:** 7
**Confidence:** 4

**Summary:**

Matrix mechanisms are a foundational class of methods in differential privacy for efficiently answering large sets of marginal queries on tabular data. However, matrix mechanisms infamously suffer in high-dimensional data domains, where memory and compute explode. Significant efforts have been made to overcome this hurdle including the HDMM, the Private-PGM, and recently, the ResidualPlanner.

This paper generalizes the residual decomposition techniques of ResidualPlanner to be far more effective and usable. They authors demonstrate how to answer arbitrary workloads of queries from arbitrary sets of residuals in a principled loss-minimizing way. They then extend this technique to offer non-negativity (when marginals cannot be negative-valued), and demonstrate how to apply it for the canonical MWEM mechanism.

The show the significantly improved error of their reconstructions over a variety of baselines including the original ResidualPlanner. Their method is only improved on by Private-PGM, which suffers heavily from memory blow-up (in my own experience as well).

**Strengths:**

The methods demonstrated in this paper constitute a significant improvement to the state of the art in a foundational problem in Differential Privacy. They take the clever concepts of residual-based high-dimensional reconstruction and make them practical across a wide range of settings. They demonstrate how these methods can be used for canonical mechanisms and a significant improvement in utility on standard datasets/tasks.

**Weaknesses:**

There is a lot of technical content in this paper, which is challenging to get through in 9 pages. The writing could have been organized differently to help the reader gain intuition on how this method improves the state of the art. It took me a few reads.

For instance, it is clear to me that the fundamental challenge is the computational intractability of the pseudoinverse, and that residual workloads help solve this. I still do not have great intuition on how residual workloads make this improvement. Perhaps I’m slow on this point, but it feels like it should be illuminated more prominently since it is central to the problem. Granted, residual workloads were introduced in a prior work.

Another confusion: a major asset of ReM when it is introduced is that a given residual can be queried multiple times and then more accurately estimated with MLE. It was not clear to me why one would want to measure a single residual multiple times. Later in the scalable MWEM section, I could see how a residual would be queried multiple times as it appears in multiple marginals, but I was not sure if that’s the only reason. I think the motivation behind ReM could be described differently, perhaps even starting with something the reader knows (MWEM) and then showing how ReM and grem-lnn come to the rescue.

**Questions:**

See two questions in weaknesses section.

**Limitations:**

I think the authors’ treatment of limitations covers most bases I would consider.

---

> ### Author Rebuttal · Authors · 2024-08-07
>
> We thank the reviewer for providing feedback on our paper and suggesting improvements to clarify the presentation.
>
> In the revision, we will do our best to improve readability of the paper and better motivate residual queries in Section 2.1.
>
> Regarding why there might be multiple measurements of the same residual, please see the global response.

---

> > ### Comment · Reviewer_a9Gz · 2024-08-13
> >
> > I'm glad to hear the paper motivation will be revised. The examples of repeated measurements of a single residual in the global response clarified things for me.

---

### Official Review · Reviewer_hTTb · 2024-07-13

**Soundness:** 3
**Presentation:** 3
**Contribution:** 3
**Rating:** 6
**Confidence:** 3

**Summary:**

The paper introduces ReM method for efficiently reconstructing answers to marginal queries with differential privacy. This aim is to minimize the error and allow scalability to high-dimensional datasets. As an extension, this paper also proposes ReM-LNN which ensures that the reconstructed marginals are non-negative, which further reduces error. The effectiveness of these methods is demonstrated by comparing them with existing private query-answering mechanisms like ResidualPlanner and MWEM.

**Strengths:**

This paper is well-written. The comparisons with prior works are adequately addressed. The proposed method is novel and practical.

**Weaknesses:**

It is not clear how to apply the reconstructed marginal queries in practical settings. For example, Private-PGM is able to generate synthetic datasets for further downstream ML tasks. However, the proposed method is not able to generate synthetic datasets.

**Questions:**

The proposed method claims to be scalable. I am wondering how the time complexity of the dual ascent algorithm used for solving (1) compares with other methods, especially how it scales with the dataset.

**Limitations:**

The limitations are adequately addressed.

---

> ### Author Rebuttal · Authors · 2024-08-07
>
> We thank the reviewer for providing feedback on our paper and suggesting improvements to clarify the presentation. Below, we address various concerns raised in the review.
>
> Private query answering is valuable in its own right outside of its application for synthetic data generation. Much work in differential privacy has focused on minimizing error for query answering while satisfying differential privacy. For example, for a fixed privacy budget and target workload of queries, matrix mechanisms minimize error --- often $L_2$ error --- with respect to the output answers. The original matrix mechanism [0] can answer arbitrary linear queries but is inefficient, HDMM [1] restricts to highly structured linear queries such as marginals and range queries and is more scalable than the original matrix mechanism, and ResidualPlanner [2] efficiently answers marginal queries. ReM is a generalization of the reconstruction step of ResidualPlanner.
>
> Thank you for your questions about time complexity. Please see the global response for time complexity results, which we will add to the paper.
>
> Since the submission of the paper, we have been working to improve the dual ascent algorithm in practice by incorporating line search over the step size $t$ in Line 6 of Algorithm 4. This reduces the number of rounds of dual ascent until convergence.
>
> [0] Li, Chao, et al. "The matrix mechanism: optimizing linear counting queries under differential privacy." The VLDB journal 24 (2015): 757-781
>
> [1] McKenna, Ryan, et al. "Optimizing Error of High-Dimensional Statistical Queries Under Differential Privacy." Journal of Privacy and Confidentiality 13.1 (2023).
>
> [2] Xiao, Yingtai, et al. "An optimal and scalable matrix mechanism for noisy marginals under convex loss functions." Advances in Neural Information Processing Systems 36 (2024).

---

> > ### Comment · Reviewer_hTTb · 2024-08-14
> >
> > Thank you for addressing my question. My score remains unchanged.

---

### Author Rebuttal · Authors · 2024-08-07

Below, we address two concerns raised in the reviews: time complexity and natural cases where multiple residuals are measured.

Regarding time complexity, let $\gamma, \tau$ be tuples of attributes and suppose each attribute has domain size $m$. Then an answer to marginal $M_\gamma$ has size $m^{|\gamma|}$. Reconstructing an answer to marginal $M_\gamma$ from a set of residuals using GReM is $\mathcal{O}(| \gamma | \cdot m^{|\gamma| + 1} \cdot 2^{|\gamma| - 1})$.
This running time is subquadratic with respect to output size $m^{|\gamma|}$.
This can be verified by observing that $\Big( \frac{| \gamma | m^{|\gamma| + 1} 2^{|\gamma| - 1}}{m^{2|\gamma|}} \Big) \rightarrow 0$ as either $m$ or $|\gamma|$ grow arbitrarily large.
As a pre-processing step, we only store one combined ``measurement'' per residual in the downward closure of the workload. After this pre-processing step, the input size is no more than the output size. Therefore, the reported running time bounds are in terms of the output size only.

For the dual ascent algorithm in GReM-LNN, one round has time complexity $\mathcal{O}(| \mathcal{W} | \cdot | \gamma^* | \cdot m^{|\gamma^*| + 1} \cdot 2^{|\gamma^*| - 1} )$ where $\mathcal{W}$ is the workload of marginals to answer and $\gamma^*$ is the largest tuple of attributes in $\mathcal{W}$.
This running time is linear with respect to the number of marginals in the marginal workload $\mathcal{W}$ and subquadratic with respect to the size of the largest reconstructed marginal $M_\gamma$.

Consider the workload of all $k$-way marginals over a data domain with $d$ attributes. Reconstructing all marginals in this workload using GReM is $\mathcal{O}(k d^k m^{k+1} 2^{k-1})$.
Since $k$ appears only as an exponent or multiplicative term, reconstruction is not exponential in any size parameter other than the size of the marginals, for which we expect exponential scaling.
Similarly, for GReM-LNN, each round of dual ascent is $\mathcal{O}(k d^k m^{k+1} 2^{k-1})$.
In many use cases, only 3-way or smaller marginals are measured.

In the revision, we will include these time complexity results in the main text as well as proofs in an Appendix.

Regarding multiple measurements of the same residual, there are several reasons we might encounter this. First, some state of the art mechanisms regularly measure the same query multiple times. A good example is AIM, which has a budget annealing process that allows it to measure queries more accurately in later rounds. Because of this it will often take repeated measurements of the same marginal at decreasing noise levels.
Another reason for measuring a residual query multiple times is the conversion from marginal measurements to residual measurements. Synthetic data mechanisms like AIM and PrivMRF select marginals to measure with isotropic Gaussian noise.
In Theorem 2, we show how to convert such a marginal measurement into an equivalent set of residual measurements. When using this conversion within such a mechanism that selects marginals, a residual will be measured each time a marginal that contains it is selected.
We will include this motivation earlier in the paper to better understand the problem setting.

---

### Decision · Program_Chairs · 2024-09-25

**Decision:**

Accept (poster)

**Comment:**

The paper proposes a more efficient method for reconstructing marginal query answers from privately released residuals. This is an important step in several private query release algorithms. Some details about running time bounds need to be clarified in the final version. Otherwise, this is an original contribution making an advance on an important problem.